# Robust Policy Adaptation in Markov Decision Processes under Environmental Shift

## Abstract

Robust Markov Decision Processes (MDPs) address environmental uncertainty through distributionally robust optimization (DRO) by finding an optimal worst-case policy within an uncertainty set of transition kernels. However, the vanilla DRO approach often proves overly conservative when facing significantly different environments, resulting in suboptimal performance. In this paper, we focus on obtaining a robust policy to address environmental shifts within the framework of two-environment MDPs: the source and target domains, characterized by their transition kernels. While the source domain MDP is predetermined, access to the target domain is limited to a few samples. Our goal is to derive a robust policy that performs well in the target domain. Underlying our approach is the construction of alternative uncertainty sets, obtained through constrained estimation. This estimation leverages available target domain data and various forms of side information from the source domain to mitigate environmental discrepancies. We demonstrate the convergence of this estimation in total variation under mild assumptions. Moreover, we establish error bounds and convergence results for both the robust and non-robust value functions. Through our analysis, we illustrate the efficacy of our method in adapting to domain shifts. We assess the performance of our approach across popular OpenAI Gym environments and real-world control problems, consistently showcasing superior performance in both robust and non-robust scenarios within the target domain.

## 1 Introduction

Markov Decision Processes (MDPs) serve as powerful frameworks for modeling decision-making under uncertainty (Derman, 1970). They have underpinned significant advancements in Reinforcement Learning (RL), a branch of machine learning focused on learning optimal behavior through interaction with an environment (Sutton & Barto, 2018; Bertsekas, 2011; Puterman, 1994), aiming to find a policy to select control actions that maximize cumulative rewards. Although MDPs have demonstrated remarkable success in addressing RL challenges across various domains, real-world decision-making scenarios often encounter environment model mismatch, where discrepancies arise between the environment where the training data is collected and the target environment where the learned policy is deployed. This issue, known as environment model mismatch, is common in RL settings due to various factors, including modeling error in simulation construction, adversarial attacks or external disturbances in the true environment, and non-stationarity of the environments. These factors result in performance degradation when directly deploying the learned policy.

One way to address the challenge of environment model mismatch is to apply a transfer learning approach. Instead of directly applying the learned policy from the training environment to the target environment, additional steps are taken to further improve performance and mitigate degradation. Among various transfer learning approaches, one promising approach is robust MDPs/RL (Bagnell et al., 2001; Nilim & El Ghaoui, 2004; Iyengar, 2005), which provides a solid theoretical foundation for the performance of the learned policy in the target environment. Unlike vanilla MDP or RL, in robust MDPs, the environment transitions follow uncertain dynamics rather than a fixed one, and this collection of possible transition dynamics is known as the uncertainty set. The agent aims to learn a policy that is robust to this uncertainty by optimizing the worst-case performance across all possible environments in the uncertainty set. Consequently, the learned

policy provides a lower-bound guarantee on performance under any environment considered. Specifically, if the target environment falls within the uncertainty set, the optimal robust policy will also guarantee performance and prevent significant degradation. However, robust-RL-based transfer learning can be overly conservative. When the environment shift or model mismatch is substantial, the uncertainty set must be enlarged sufficiently to ensure inclusion, which generally leads to a pessimistic worst-case scenario. This enlargement can cause the robust policy to underperform in the target environment.

Other notable research efforts aimed at improving policy performance in related environments focus on multi-task learning (Huang et al., 2022; Du et al., 2021). These methods primarily rely on the assumption of linear MDPs and aim to learn a common feature representation to estimate a policy that performs well across all tasks. Multi-task learning enhances policy performance in the target domain by learning shared representations across multiple tasks, which can be transferred to improve performance on the target task. This is typically achieved through joint training on multiple tasks or by learning features that generalize across tasks (Cheng et al., 2022; Agarwal et al., 2023). Departing from representational learning and instead, exploiting a pre-trained policy in the training environment, to adapt it to the testing environment through a technique known as policy adaptation (Song et al., 2020). Domain randomization, which diversifies the training sample by pre-training a robust policy to a predefined distribution, can complement policy adaptation (Peng et al., 2018; Tobin et al., 2017). Other methods use meta-learning to adapt policies to perturbed environments (Finn et al., 2017; Nagabandi et al., 2018). Additional approaches employ model-free methods to learn a target domain policy. For instance, (Laroche & Barlier, 2017) utilizes transition samples learned from a specific task for another, while (Tirinzoni et al., 2018) employs an instance-weighting approach to re-weight the source domain samples. Despite the success of these methods in transfer learning, a common shortcoming is their potential inadequacy if the target environment is perturbed, as they fail to account for the inherent uncertainty in the target environment.

In this paper, we develop a framework to address the issues arising from significant environment shifts. Specifically, we propose incorporating additional prior information or domain knowledge regarding the relationship between the two environments to achieve a more accurate estimate of the target environment. This approach aims to reduce performance degradation and prevent overly conservative behavior in previous transfer learning methods like robust RL. Unlike previous robust-RL-based transfer learning approaches that do not extensively utilize side information about the source and target environments, our approach is expected to provide a more accurate estimation and yield a better policy for the target environment. We will specifically consider two approaches: non-robust vanilla RL, which directly learns the optimal policy under the estimated model, and robust RL, which learns a robust policy with respect to some uncertainty around this estimation. Our contributions are summarized as follows.

**Summary of Contributions:**

1. **We propose an Information-Based (IB) approach for estimating the target transition kernels while leveraging side information from the source MDP.** For the target domain MDP, we utilize the limited number of samples available and incorporate prior information about its relationship with the source MDP. By translating varying degrees of prior information into constraints that the target environment should satisfy, we develop an IB formulation to estimate the transition kernel of the target domain. We establish constrained Cramér–Rao bounds, showing that a more robust estimator can be achieved when stronger prior information is available regarding the relationship between the two domains, resulting in a better lower bound on the variance of the estimator. Our experimental results highlight the efficiency of our prior-constrained estimator, demonstrating its ability to capture more information about the kernels and produce a more accurate estimation compared to its unconstrained counterpart.

2. **We establish the asymptotic convergence of our constrained IB estimators.** We demonstrate that in the constrained setting, the IB estimator converges asymptotically to the true target domain transition kernel in terms of total variation distance. This result is built upon the consistency of the unconstrained Maximum Likelihood Estimator (MLE), combined with the finiteness of the state space and the convexity of the side information constraints. This theoretical result is further supported by

experiments conducted across various environments, showing the asymptotic convergence of all our estimators to the target kernel.

3. **We derive error bounds and convergence results for the value functions in both robust and non-robust settings.** In our main result, we establish asymptotic performance guarantees for the policies learned under our models, demonstrating that our methods yield a near-optimal policy in the target domain. Specifically, we prove that the sub-optimality gap between our policies and the optimal policy in the target domain depends solely on the total variation distance between the estimated and true target kernels. Combined with our previous results, we further show the convergence of our policies to the optimal policy. Additionally, we extend our results to the robust setting, where we show that our framework can also learn a near-optimal robust policy accounting for model uncertainty in the target domain. These theoretical results are further validated through experimental verification, illustrating the asymptotic convergence of the expected reward function to the target domain.

4. **We validate our approach through experiments on environments from OpenAI Gym and control problems.** We evaluate the performance of our approach across six environments from OpenAI Gym, including both toy-text scenarios (frozen lake, cliff walking, and taxi) and control problems (cart pole, acrobot, and pendulum). Our experimental results consistently outperform state-of-the-art methods, which underscores the efficacy of incorporating prior information in enhancing performance.

The paper is organized as follows. Section 2 discusses related work. Section 3 presents preliminary background and outlines the problem formulation. Our main approach is detailed in Section 4, where we describe and analyze our method for estimating the transition kernels and learning the target policy. We also establish our main theoretical guarantees on convergence under both non-robust and robust settings. Numerical experiments on various OpenAI Gym environments are provided in Section 5. Appendix A contains a table summarizing the main notation used in this work. The proofs of the main and intermediate results are deferred to Appendix B. Details of the algorithms, environment descriptions, and additional experimental results are presented in Appendices C, D, and E, respectively.

## 2 Related Work

**Robust RL.** Robust RL has attracted attention in recent years due to its potential to identify policies that remain effective across uncertain environments. Recent research has adopted a distributionally robust optimization (DRO) framework to enhance robustness by optimizing the worst-case performance over an uncertainty set of transition kernels or environments (Iyengar, 2005; Nilim & El Ghaoui, 2004; Bagnell et al., 2001; Wiesemann et al., 2013; Lim et al., 2013; Tamar et al., 2014). The methods for solving robust MDPs can be classified into two classes: model-based and model-free. Both classes aim to optimize the worst-case performance given samples generated from a single training environment. In model-based approaches, an estimate of the MDP model is first obtained, based on which robust dynamic programming is applied (Lim & Autef, 2019; Yu & Xu, 2015; Yang et al., 2021; Shi et al., 2023; Panaganti & Kalathil, 2021; Panaganti et al., 2022; Wang et al., 2023b); while in model-free settings, the agent aims to learn the optimal policy without estimating the model (Roy et al., 2017; Badrinath & Kalathil, 2021; Wang & Zou, 2021; Tessler et al., 2019; Zhou et al., 2021; Goyal & Grand-Clement, 2018; Kaufman & Schaefer, 2013; Ho et al., 2018; 2021; Si et al., 2020; Xu & Mannor, 2010; Wang et al., 2024a; Liu et al., 2022; Wang et al., 2023a;c). This approach can provide performance guarantees in the target environment if the uncertainty set includes the target one. While the worst-case performance provides a lower bound on the reward in the test environment—assuming the testing kernel is contained within the uncertainty set—a notable limitation arises when the testing (target) domain transition kernel diverges significantly from the training environment (source domain). In such scenarios, a large uncertainty set is required, which tends to yield an overly conservative, unduly regularized solution, leading to poor performance.

**Multi-Task RL.** Another related research direction is multi-task RL, where the goal is to find a set of actions that perform well across multiple tasks (environments) (Huang et al., 2022; Du et al., 2021). Most

existing approaches assume that the MDP is linear or of low rank, focusing on extracting a common feature representation for all tasks. In contrast, transfer learning in RL aims to enhance performance in the target domain specifically. Some works have proposed multi-task transfer learning algorithms that primarily focus on target domain environments. Cheng et al. (Cheng et al., 2022) conduct reward-free exploration within source tasks to refine representation learning, utilizing the acquired representation for online learning in the target task. However, they impose strong assumptions for transfer, such as a linear span with fixed coefficients, transition dynamics error bounded by expected value error, and reachability in high-dimensional space. These assumptions were relaxed in recent work by Agarwal et al. (Agarwal et al., 2023), which considers a general state-dependent linear assumption and uses in-distribution generalization with a novel cross-sampling approach. Our work, however, differs in its setting; we assume a limited number of independent samples are available from the target domain, whereas their approach assumes the agent can interact with the target environment during the deployment phase (online adaptation).

**Policy Adaptation.** Some research focuses on transferring a pre-trained policy to another domain, a process known as policy adaptation. Some approaches use domain randomization to adapt the pre-trained policy to the target domain, relying on the assumption that both domains' environments are drawn from the same predefined training distribution (Peng et al., 2018; Tobin et al., 2017; Chebotar et al., 2019). Other policy adaptation methods use meta-learning to learn a policy on variety of environments to adapt to the target domain (Finn et al., 2017; Nagabandi et al., 2018). However, most of these methods assume knowledge of the true target dynamics, making them available for sampling during the training phase, and often perform poorly on out-of-distribution target domains or relatively distant environments. Song et al. (Song et al., 2020) propose a model-based approach for policy adaptation, alternating between two steps: a modeling step that learns the divergence between the environments' dynamics, and a planning step that determines actions to minimize this divergence. Their method primarily relies on online adaptation, using data aggregation to gather information as the policy is updated.

**DA Techniques.** Another line of research, emerging from the expanding literature on domain adaptation (DA), focuses on harnessing transfer learning techniques to enhance performance in the target domain by reducing the disparity between domains. Taylor et al. (Taylor et al., 2008) proposed an approach to transfer samples from the source to the target task using inter-task mapping, while Laroche et al. (Laroche & Barlier, 2017) introduced a technique for transferring samples using a model-free approach called Fitted Q-Iteration (FQI). However, both works assume common transition dynamics across the two domains, a limitation that differs from our problem setup. In our work, we consider an altogether different setting, addressing the mismatch between the transition kernels of the two domains. This allows us to mitigate negative transfer by obtaining a robust estimate of the target domain kernels, which is particularly beneficial in scenarios with relatively small sample sizes.

Tirinzoni et al. (Tirinzoni et al., 2018) adopt the FQI approach and utilize importance weighting to select the most relevant source samples for transfer. Their method assumes multiple source domains and one target domain, where both the transition dynamics and reward function differ. Gaussian Processes (GPs) are employed to estimate the weights. However, a primary limitation of this approach is the assumption that the transition kernels are Gaussian, upon which the weights are calculated. Consequently, this method is unsuitable for discrete state spaces and is limited to those following Gaussian dynamics. In contrast, our approach makes no assumptions about the estimated transition kernels, making it applicable to different types of distributions.

**Robust DA in Supervised Learning.** In the realm of supervised learning, numerous research efforts have addressed the DA problem. The primary objective of DA is to estimate a decision model that performs effectively on the target domain by mitigating the discrepancy between domains resulting from distribution mismatch (Weiss et al., 2016). Various techniques have been employed to achieve this, including instance weighting (Sugiyama et al., 2007; Huang et al., 2006) and feature alignment (Long et al., 2013; Zhang et al., 2017; Alkhouri et al., 2023), among others. However, most existing DA methods overlook the uncertainty inherent in the distributions, limiting their effectiveness in noisy environments or with limited sample sizes. Recent works have attempted to address this limitation by incorporating the twin objectives of performing well on the target domain and being robust to uncertainty in the target domain distribution (Liu & Ziebart, 2014; Chen et al., 2016; Awad & Atia, 2023; Wang & Wang, 2024). Nevertheless, to the best of our knowledge,

this has not yet been explored in the context of policy adaptation for decision-making. In this paper, we present a novel approach to DA under environment shift, involving source and target MDPs with differing transition kernels. In the non-robust setting, our primary objective is to estimate a policy that performs effectively in the target domain. In the robust setting, we augment this objective with the aim of achieving robustness against a set of transition kernels, thereby accounting for uncertainty in the target environment.

## 3 Preliminaries and Problem Formulation

**Markov Decision Process (MDP).** A MDP is denoted by the tuple $(\mathcal{S}, \mathcal{A}, \mathsf{P}, r, \gamma)$, where $\mathcal{S}$ is the state space, $\mathcal{A}$ is the action space, $0 \leq \gamma < 1$ is a discount factor, $r : \mathcal{S} \times \mathcal{A} \to \mathbb{R}$ is a reward function mapping state-action pairs to a real reward value, and transition kernels $\mathsf{P} = \{\mathsf{P}^{s,a} \in \Delta(\mathcal{S}), a \in \mathcal{A}, s \in \mathcal{S}\}$[1], where $\mathsf{P}^{s,a}$ is a probability mass function (PMF) over the state space $\mathcal{S}$ of the next state conditioned on state $s$ and action $a$. Hence, $\mathsf{P}^{s,a}(s')$ denotes the probability of arriving at state $s' \in \mathcal{S}$ if the agent is in state $s$ and takes action $a$. At each time step $t \geq 0$, the environment transits to the state $s_{t+1}$ according to the transition probability $\mathsf{P}^{s_t,a_t}(s_{t+1})$ and yields a reward $r(s_t, a_t)$ for the agent.

A stationary policy $\pi : \mathcal{S} \to \Delta(\mathcal{A})$ is a distribution over the set of actions $\mathcal{A}$ for a given state $s$, which determines the probability of selecting a given action at a certain state. The value function of a stationary policy $\pi$ at state $s$ is defined as the expected discounted cumulative reward if the agent starts from state $s$ and takes actions according to policy $\pi$, i.e.,

$$V_{\mathsf{P}}^{\pi}(s) = \mathbb{E}_{\pi,\mathsf{P}} \left[ \sum_{t=0}^{\infty} \gamma^t r(s_t, a_t) \big| S_0 = s \right]. \tag{1}$$

The goal of the agent is to learn a stationary policy to maximize the expected cumulative reward, i.e.,

$$\pi^* = \arg\max_{\pi} V_{\mathsf{P}}^{\pi}. \tag{2}$$

**Robust MDP.** The robust MDP is defined by the tuple $(\mathcal{S}, \mathcal{A}, \mathcal{P}, r, \gamma)$, where the transition kernel is not fixed but comes from an uncertainty set $\mathcal{P}$. The environment can transit to the next state according to an arbitrary transition kernel belonging to the uncertainty set. In this work, we consider the $(s, a)$-rectangular uncertainty set (Nilim & El Ghaoui, 2004; Iyengar, 2005), i.e., $\mathcal{P} = \bigotimes_{s,a} \mathcal{P}_s^a$, where $\mathcal{P}_s^a \subseteq \Delta(\mathcal{S})$ and $\bigotimes_{s,a}$ is the cross-product over all state-action pairs. The robust discounted value function of policy $\pi$ at state $s$ is defined as

$$V_{\mathcal{P}}^{\pi}(s) \triangleq \min_{\eta = (\mathsf{P}_0, \mathsf{P}_1, \dots) \in \bigotimes_{t \geq 0} \mathcal{P}} \mathbb{E}_{\pi,\eta} \left[ \sum_{t=0}^{\infty} \gamma^t r(s_t, a_t) | S_0 = s \right], \tag{3}$$

which accounts for the worst-case performance over the uncertainty set of transition kernels. The goal of robust MDP is to optimize the worst-case performance, by maximizing the robust value function:

$$\pi^* = \arg\max_{\pi} V_{\mathcal{P}}^{\pi}. \tag{4}$$

Equation 4 finds the optimal policy that maximizes the worst-case value function, over the uncertainty set of distributions $\mathcal{P}$. The algorithms used to solve the MDPs in the robust and non-robust settings are presented in Appendix C.

**Environment Shift.** An environment shift refers to the scenario where an agent is trained in one (source) environment and deployed in a related but different (target) environment. Formally, the source domain MDP is defined as $(\mathcal{S}, \mathcal{A}, \mathsf{P}_\mathsf{s}, r, \gamma)$, and the target domain MDP is $(\mathcal{S}', \mathcal{A}', \mathsf{P}_\mathsf{t}, r', \gamma)$. In this work, we assume the two MDPs differ only in their transition kernels, and $\mathcal{S} = \mathcal{S}'$, $\mathcal{A} = \mathcal{A}'$, $r = r'$. We assume direct access to the source MDP, while only a limited dataset generated from the target MDP is available.

---

[1] The notation $\Delta(\mathcal{S})$ denotes the probability simplex over $\mathcal{S}$.

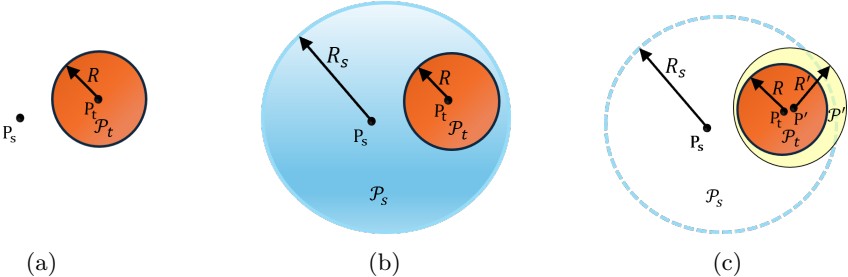

Figure 1: **(a) Environment shift:** The source and target domain environments are relatively distant. **(b) Over-conservative case:** the source uncertainty set's radius is enlarged to include the target domain, which leads to an overly conservative policy. **(c) Our approach**: We construct an uncertainty set around the estimated target domain transition kernel, which is derived from limited target domain samples and source side information.

**Problem Formulation.** We are given a source domain MDP $(\mathcal{S}, \mathcal{A}, \mathsf{P_s}, r, \gamma)$ and i.i.d. samples $\{(s_i, a_i, s_i')\}_{i=1}^N$, from the target MDP $(\mathcal{S}, \mathcal{A}, \mathsf{P_t}, r, \gamma)$, consisting of current states $s_i \in \mathcal{S}$, actions $a_i \in \mathcal{A}$, and future states $s_i' \sim \mathsf{P}_t^{s_i, a_i}$. Also, we assume the availability of side information, denoted $\Phi(\mathsf{P_s}, \mathsf{P_t})$, about the relationship between the transition kernels of both domains (see Section 4.2.1, where we consider various forms of side information). Our goal is two-fold: first, to learn an optimal policy for the target MDP; second, to find an optimal robust policy that accounts for model uncertainty in the target environment.

# 4 Main Approach: Learning Policy Under Environment Shift

## 4.1 Motivation

The primary challenge in our problem is that the target environment is only observable through a limited number of samples. A straightforward approach is to learn a policy directly from the target domain data using offline RL algorithms, such as (Uehara & Sun, 2021; Wang et al., 2024b). However, directly applying offline methods requires a large number of high-quality samples to obtain an optimal target domain policy, which may not be feasible in a limited data setting. Specifically, when the sample size is very limited, offline RL approaches tend to produce unsatisfactory policies, as demonstrated in previous offline RL studies (Levine et al., 2020)

Alternatively, one could leverage the source MDP to learn an optimal policy and directly deploy it in the target environment. While this might yield good performance if the transition kernels of the two MDPs are similar, the performance can significantly deteriorate as the discrepancy between the two domains increases.

Previous works have turned to robust MDPs, optimizing over a set of distributions centered around the source transition kernels including the target one, to find a policy that maximizes the worst-case performance and to provide an optimized lower bound on its performance in the target environment (Wang et al., 2023b; Wiesemann et al., 2013; Tamar et al., 2014; Lim & Autef, 2019; Xu & Mannor, 2010; Yu & Xu, 2015; Lim et al., 2013). However, as mentioned earlier, this approach may yield an overly pessimistic worst-case performance and perform poorly in the target environment, especially when the environmental shift is large and a large uncertainty set is constructed to include the target transition kernel.

To address these challenges, we propose a method to transfer knowledge from the source to the target domain by leveraging information from both the source MDP and the limited target domain samples. This approach aims to improve the performance of the learned policy in the target domain. We optimize performance both without and with environmental uncertainty by: (1) learning the optimal policy for the target domain, and (2) learning the optimal robust policy for an uncertainty set centered around the target environment.

In this paper, we focus on model-based approaches. Our approach involves three key steps: 1) estimating the target domain transition kernels, 2) estimating the policy, and 3) evaluating the policy in the target domain environment. We assume the availability of prior information about the relationship between the source and target transition kernels in various forms (See Sec. 4.2), which is shown to enhance the learned

policy's performance significantly. A visualization of the environment shift problem, over-conservative case, and our approach are depicted in Figure 1. The environment shift is presented in Figure 1a where the source and target transition kernels, $\mathsf{P}_s$ and $\mathsf{P}_t$, respectively, are distant. The goal is to learn a policy that performs well in the target environment. The policy is evaluated on the target domain uncertainty set $\mathcal{P}_t$ of radius $R \geq 0$. Figure 1b shows the over-conservative case in which an uncertainty set $\mathcal{P}_s$ with radius $R_s \geq R$, is constructed around the source transition kernels. For our approach, presented in Figure 1c, we find the policy by optimizing over an uncertainty set $\mathcal{P}'$ of radius $R'$ centered around estimated target domain transition kernels $\mathsf{P}'$. It is important to note that our approach encompasses both the robust and non-robust cases: in the non-robust case, the training and evaluation uncertainty sets are singletons, i.e., $R = R' = 0$.

**Remark 1.** *The uncertainty set centered around the estimated kernels requires a smaller radius to include* $\mathsf{P}_t$ *compared to the over-conservative case where the uncertainty set is centered around the source distribution. This uncertainty set, defined with respect to TV distance, is then used to obtain the policy. For example, consider the uncertainty set* $\mathcal{P}'$ *defined with respect to the TV distance with radius* $R'$ *and centered at the estimated kernel* $\mathsf{P}'$. *Let* $\pi^*$ *denote the estimated policy such that* $TV(\mathsf{P}_t, \mathsf{P}') \leq R'$. *Then, it is guaranteed that the value function satisfies:*

$$V_{\mathsf{P}_t}^{\pi^*}(s) \geq V_{\mathcal{P}}^{\pi^*}(s), \quad \forall s \in \mathcal{S},$$

*which provides a lower bound on the value function evaluated at the true* $\mathsf{P}_t$. *Thus, using the TV metric offers insight into the quality of the policy, since it is the same metric used in defining the uncertainty set.*

## 4.2 Information-Based Estimation Approach

As discussed above, our goal is to obtain a more accurate estimate of the target transition kernel. In this section, we propose our estimation approach and show that it is more sample-efficient.

### 4.2.1 Estimation with side information

In the context of transfer learning, we assume the availability of prior information about the relationship between the source and target environments, specifically between their transition kernels. We are given prior information on $\mathsf{P}_t$, denoted as $\Phi(\mathsf{P}_t, \mathsf{P}_s)$, which captures specific constraints representing the similarity or relationship between the source and target transition kernels. We propose to utilize a Constrained Maximum Likelihood Estimator (CMLE) to estimate the target domain distribution. Given $n$ i.i.d. samples drawn from the target domain transition kernel $\mathsf{P}_t^{s,a}$, we denote the estimate of $\mathsf{P}_t^{s,a}$ by $\hat{\mathsf{P}}_n^{s,a}$. Hence, we seek to solve the following general program:

$$\max_{\mathsf{P}^{s,a} \in \Delta(\mathcal{S})} \sum_{j=1}^{|\mathcal{S}|} n_j \log \mathsf{P}_j^{s,a} \quad \text{s.t.} \quad \Phi(\mathsf{P}^{s,a}, \mathsf{P}_s^{s,a}), \tag{5}$$

where $n_j = \sum_{i=1}^n \mathbb{I}[s_i' = j]$ represents the number of samples among the $n$ total samples such that the next state $s' = j$, where $\mathbb{I}[\cdot]$ is the indicator function, and $\sum_{j=1}^K n_j = n$. $\mathsf{P}_j^{s,a}$ represents the entry of $\mathsf{P}^{s,a}$ corresponding to state $s_j \in \mathcal{S}$. We refer to this estimation method as Information-Based (IB) estimation approach, and to the estimator $\hat{\mathsf{P}}_n^{s,a}$ obtained by solving the program in (5) as the Information Based Estimator (IBE). We consider different types of side information, namely, distance, moments, and density ratio constraints, and choose the appropriate method based on the constraint $\Phi(\mathsf{P}^{s,a}, \mathsf{P}_s^{s,a})$ as described below.

**Vanilla IBE:** Used as a baseline, this is an offline RL approach which estimates the transition kernel using the offline data, without accounting for any side information.

**Distance IBE:** Here, we have an upper bound on the distance between the kernels. Specifically, the $l_1$-norm distance between $\mathsf{P}_t^{s,a}$ and $\mathsf{P}_s^{s,a}$ is assumed to be at most $d$, i.e., $\|\mathsf{P}_t^{s,a} - \mathsf{P}_s^{s,a}\|_1 \leq d$.

**Moment IBE:** In this case, we have prior information in the form of moment constraints. The embedding mean of the probability distribution $\mathsf{P}^{s,a}$ is defined as a high-dimensional moment vector $\mu_{\mathsf{P}^{s,a}} = \mathbb{E}_{\mathsf{P}^{s,a}}[\phi]$, where $\phi : \mathcal{S} \to \mathcal{H}$ maps the state space to a Hilbert space. We assume that the vector of moments of the source is at most at distance $\delta$ from the target moments, i.e., $|\mu_{\mathsf{P}_t^{s,a}} - \mu_{\mathsf{P}_s^{s,a}}| \leq \delta$.

Table 1: Different IB estimation methods for different prior information.

| VANILLA IBE | DISTANCE IBE | DENSITY IBE | MOMENT IBE |
|---|---|---|---|
| $\max_{P \in \Delta(S)} \sum_{j=1}^{k} n_j \log(\mathsf{P}_j^{s,a})$ | $\max_{P \in \Delta(S)} \sum_{j=1}^{k} n_j \log(\mathsf{P}_j^{s,a})$ $\|\mathsf{P}_{\mathrm{s}}^{s,a} - \mathsf{P}^{s,a}\|_1 \leq d$ | $\max_{P \in \Delta(S)} \sum_{j=1}^{k} n_j \log(\mathsf{P}_j^{s,a})$ $\mathsf{P}^{s,a}/\mathsf{P}_{\mathrm{s}}^{s,a} \leq B$ | $\max_{P \in \Delta(S)} \sum_{j=1}^{k} n_j \log(\mathsf{P}_j^{s,a})$ $|\mu_{\mathsf{P}^{s,a}} - \mu_{\mathsf{P}_{\mathrm{s}}^{s,a}}| \leq \delta.$ |

**Density IBE:** In this setting, we assume prior information on the density ratio between $\mathsf{P}_{\mathrm{t}}^{s,a}$ and $\mathsf{P}_{\mathrm{s}}^{s,a}$. We define the density ratio $w(s) = \frac{\mathsf{P}_{\mathrm{t}}^{s,a}(s)}{\mathsf{P}_{\mathrm{s}}^{s,a}(s)}$, assuming that the target distribution $\mathsf{P}_{\mathrm{t}}^{s,a}$ is absolutely continuous w.r.t. the source distribution $\mathsf{P}_{\mathrm{s}}^{s,a}(s)$. Further, we assume that the density ratio is upper bounded by $B$.

Table 1 summarizes our formulated programs for the different cases considered above. Next, we state our results on the convergence of the estimators proposed in this section.

**Theorem 1.** *Let $(\hat{\mathsf{P}}_n^{s,a})^*$ be a sequence of transition kernels obtained as the solution to (5), with the constraint sets presented in Table 1, where $n$ is the sample size. Further, assume that the true target kernel $\mathsf{P}_{\mathrm{t}}^{s,a}$ is an interior point of the constraint set. Then, $(\hat{\mathsf{P}}_n^{s,a})^*$ converges in total variation, i.e.,*

$$\lim_{n \to \infty} \|(\hat{\mathsf{P}}_n^{s,a})^* - \mathsf{P}_{\mathrm{t}}^{s,a}\|_{TV} = 0. \tag{6}$$

Theorem 1 establishes the convergence of the IBE in total variation distance, validating our framework for estimating the target environment dynamics. More importantly, as will be shown in Section 4.3, this result further implies the effectiveness in transfer learning. In particular, it enables the learning of a policy with improved performance in the target domain.

**Remark 2.** *The side information used in this work is in the form of bounds on the TV distance, density ratios, or moment constraints between the source and target transition kernels. These types of information can be realistic in certain practical scenarios. For example, bounds on TV distance can arise naturally in settings with limited domain shifts. For instance, in robotics or control tasks, changes in dynamics such as friction or sensor noise might have quantifiable limits derived from domain knowledge or prior experiments. Density ratio information can be inferred or approximated using a small amount of data from the target domain. This is common in transfer learning applications where target domain data is scarce, but sufficient to estimate relative probabilities of transitions between the domains. Techniques like importance sampling often rely on such ratios. Moment constraints are a common way to encode known properties of the environment. For instance, in physical systems, aggregate or average behaviors are often measurable, even if the full distribution is unknown. The flexibility of our approach allows it to be adapted to a wide range of scenarios where partial knowledge about the relationship between domains is available, which could depend on the application context.*

### 4.2.2 Effect of estimation constraints on the Cramér–Rao Bound

In this section, we examine the Cramér–Rao Bound (CRB) for the estimator $\hat{\mathsf{P}}_n^{s,a}$ under the constrained estimation setting. The CRB provides a lower bound on the variance of a single parameter estimator or the covariance matrix of a vector with multiple parameters. Given an unbiased estimator $\hat{\theta}$ of a $k$-element vector $\theta$, the Fisher Information Matrix (FIM) is defined as $J(\theta) = \mathbb{E}[\nabla \log(L(\theta))\nabla \log(L(\theta))^{\top}]$, where $\nabla \log(L(\theta))$ is the gradient of the log-likelihood function w.r.t. $\theta$. The CRB then states that the covariance of the unbiased estimator is lower bounded by the inverse $J(\theta)^{-1}$ of the FIM. Before presenting the main result of this subsection, we derive an expression for the FIM $J(\mathsf{P}_{\mathrm{t}}^{s,a})$ as stated in the next lemma.

**Lemma 2.** *Given $n$ i.i.d. samples $\{x_1, \ldots, x_n\} \sim \mathsf{P}_{\mathrm{t}}^{s,a}$, where $x \in \{s_1, \ldots, s_k\}$, with $\mathsf{P}_{\mathrm{t},j}^{s,a} := \Pr(x = s_j), \mathsf{P}_{\mathrm{t},j}^{s,a} \neq 0, \forall j \leq k$, the FIM is given by*

$$J(\mathsf{P}_{\mathrm{t}}^{s,a}) = n \operatorname{diag}\left(\frac{1}{\mathsf{P}_{\mathrm{t},1}^{s,a}}, \ldots, \frac{1}{\mathsf{P}_{\mathrm{t},k}^{s,a}}\right). \tag{7}$$

We can readily state the main result of this subsection.

**Theorem 3.** *Let $\hat{\mathsf{P}}_n^{s,a}$ be an unbiased estimator of $\mathsf{P}_{\mathrm{t}}^{s,a}$ based on $n$ i.i.d. samples. For each of the following constraints, the diagonal elements of the CRB matrix, $C_{ii}(\mathsf{P}_{\mathrm{t}}^{s,a})$, are given by:*

- **_Regular Probability Constraint_** *(* $\sum_i^k \mathsf{P}_{t,i}^{s,a} = 1$ *):* $\quad C_{ii}(\mathsf{P}_t^{s,a}) = \frac{1}{n}\mathsf{P}_{t,i}^{s,a}(1 - \mathsf{P}_{t,i}^{s,a})$

- **_Moment constraints_** *(* $\mathbb{E}_{\mathsf{P}_t^{s,a}}[\phi(x)] = \mu$, *where* $x \in \mathcal{S}$ *and* $\phi : \mathcal{S} \to \mathbb{R}^m$, *and* $\sum_i^k \mathsf{P}_{t,i}^{s,a} = 1$ *) :*

$$C_{ii}(\mathsf{P}_t^{s,a}) = C_{ii}^R(\mathsf{P}_t^{s,a}) - \Delta_i \tag{8}$$

*where*

- $C_{ii}^R$ *is the* $i^{th}$ *diagonal element of the regular probability constraints CRB matrix.*
- $\Delta_i = \frac{(\mathsf{P}_{t,i}^{s,a})^2}{n^2}(a_i - \overline{a})\mathbb{S}^{-1}(a_i - \overline{a})$, *where* $a_i = \phi(x_i) - \phi(x_k) \in \mathbb{R}^m, i = 1, \ldots, k, \overline{a} = \sum_{i=1}^k a_i \mathsf{P}_{t,i}^{s,a}$, *and* $\mathbb{S} = \frac{1}{n}Cov(a)$.

Theorem 3 provides valuable insights into the quality of the estimator as additional constraints are imposed. Specifically, the diagonal elements of the CRB give lower bounds on the variances of $\hat{\mathsf{P}}_{n,i}^{s,a}, 1 \leq i \leq k$, where $\hat{\mathsf{P}}_{i,n}^{s,a}$ is the $i^{th}$ element of $\hat{\mathsf{P}}_n^{s,a}$. A smaller lower bound indicates the potential for a better minimum variance estimator. To quantify the CRB matrix, we can compute the trace of the CRB $C$, which provides a lower bound on the sum of the estimator variances. Formally, we have $\sum_{i=1}^k \mathrm{var}[\hat{\mathsf{P}}_n^{s,a}] \geq \mathrm{trace}(C(\mathsf{P}_t^{s,a})) = \sum_{i=1}^k C_{ii}(\mathsf{P}_t^{s,a})$. According to Theorem 3, we conclude that:

$$\frac{1}{n}\sum_{i=1}^k \mathsf{P}_{t,i}^{s,a}(1 - \mathsf{P}_{t,i}^{s,a}) \geq \sum_{i=1}^k \frac{\mathsf{P}_{t,i}^{s,a}(1 - \mathsf{P}_{t,i}^{s,a})}{n} - \frac{(\mathsf{P}_{t,i}^{s,a})^2(a_i - \overline{a})^\top \mathbb{S}(a_i - \overline{a})}{n^2}. \tag{9}$$

Since $\mathbb{S}$ is positive definite (assuming $Cov(a)$ is full rank), $\Delta_i \geq 0$.

Eq. 9 demonstrates that having more prior information about $\mathsf{P}_t^{s,a}$ potentially leads to a better estimate, given the smaller bound on the variance. We also provide empirical evidence of the information gain that the side information provides in Appendix E.

## 4.3 Effectiveness of Transfer Learning with Information-Based Estimation

In this section, we establish theoretical guarantees that the policy learned through our framework converges to the optimal policy in the target environment, thereby demonstrating the effectiveness of our framework. We establish convergence results for both the non-robust and robust settings.

Given a sample of size $n$ drawn i.i.d. from each $(s, a)$ pair target transition kernel $\mathsf{P}_t^{s,a}$, i.e., $\{x_i\}_{i=1}^n \overset{\text{i.i.d.}}{\sim} \mathsf{P}_t^{s,a}$, with $x_i \in \mathcal{S}$, let $\hat{\mathsf{P}}_n^{s,a}$ be the estimate of the target transition kernel. Let $V_{\mathcal{P}}^*$ denote the worst-case (robust) value function under the optimal policy $\pi^*$, over the uncertainty set $\mathcal{P} = \bigotimes_{s,a} \mathcal{P}_s^a$, where $\mathcal{P}_s^a = \{Q \in \Delta(\mathcal{S}) : \|\mathsf{P}_t^{s,a} - Q\|_{TV} \leq R\}$. For each uncertainty set $\mathcal{P}_n = \bigotimes_{s,a}(\mathcal{P}_n)_s^a$, where $(\mathcal{P}_n)_s^a = \{Q \in \Delta(\mathcal{S}) : \|\hat{\mathsf{P}}_n^{s,a} - Q\|_{TV} \leq R\}$, let the optimal policy be $\pi_n$. This uncertainty set, estimated for a sample size $n$, consists of all distributions that are within a specified distance from the estimated kernel, with a cross-product structure over the kernels for all $(s, a)$ pairs. Also, define $V_{\mathcal{P}_n}^{\pi_n}$ and $V_{\mathcal{P}}^{\pi_n}$ as the worst-case value functions under the policy $\pi_n$, over the uncertainty sets $\mathcal{P}$ and $\mathcal{P}_n$, respectively. We prove that, if $\hat{\mathsf{P}}_n^{s,a}$ converges to $\mathsf{P}_t^{s,a}$ in total variation distance, then the optimal robust (non-robust) value function under the estimated policy $\pi_n$ converges to the optimal target domain robust (non-robust) value function.

First, we establish a theorem that provides a bound on the error between $V_{\mathcal{P}}^*$ and $V_{\mathcal{P}_n}^{\pi_n}$, which we refer to as the *training error*.

**Theorem 4.** *Let* $\hat{\mathsf{P}}_n^{s,a}$ *be a sequence of transition kernels, and consider the uncertainty set* $\mathcal{P}_s^a = \{Q \in \Delta(\mathcal{S}) : \|\mathsf{P}_t^{s,a} - Q\|_{TV} \leq R\}$, *where* $\mathsf{P}_t^{s,a}$ *is the true transition kernel. Denote the optimal robust value function by* $V_{\mathcal{P}}^*$. *For each uncertainty set* $(\mathcal{P}_n)_s^a = \{Q \in \Delta(\mathcal{S}) : \|\hat{\mathsf{P}}_n^{s,a} - Q\|_{TV} \leq R\}$, *denote the optimal robust policy by* $\pi_n$. *Then, we have*

$$\|V_{\mathcal{P}_n}^{\pi_n} - V_{\mathcal{P}}^*\|_{TV} \leq \max_{(s,a)} \frac{\|\hat{\mathsf{P}}_n^{s,a} - \mathsf{P}_t^{s,a}\|_{TV}}{(1 - \gamma)^2}, \tag{10}$$

*where* $\mathcal{P} = \bigotimes_{s,a} \mathcal{P}_s^a$ *and* $\mathcal{P}_n = \bigotimes_{s,a}(\mathcal{P}_n)_s^a$.

Therefore, the training error is upper bounded by the maximum total variation distance between the two transition kernels among all $(s, a)$ pairs, indicating that the bound is dependent on how accurately we estimate $\mathsf{P}_t^{s,a}$. In other words, as the estimate $\hat{\mathsf{P}}_n^{s,a}$ improves, the upper bound shrinks, and the value function $V_{\mathcal{P}_n}^{\pi_n}$ during the training phase approaches the optimal value function $V_{\mathcal{P}}^*$. However, what about the *evaluation error* between $V_{\mathcal{P}}^{\pi_n}$ and $V_{\mathcal{P}}^*$ ? We answer this question by establishing the following results.

**Theorem 5.** *Under the same setting of Theorem 4, it holds that*

$$\|V_{\mathcal{P}}^{\pi_n} - V_{\mathcal{P}}^*\|_{TV} \leq 2 \max_{(s,a)} \frac{\|\hat{\mathsf{P}}_n^{s,a} - \mathsf{P}_t^{s,a}\|_{TV}}{(1-\gamma)^2} \ . \tag{11}$$

Theorem 5 establishes that the evaluation error is at most twice the bound on the training error. Moreover, it provides a guarantee on the performance of the policy $\pi_n$ in the target environment as $\hat{\mathsf{P}}_n^{s,a}$ approaches $\mathsf{P}_t^{s,a}$.

**Theorem 6.** *If a sequence of transition kernel $\hat{\mathsf{P}}_n^{s,a}$ converges to $\mathsf{P}_t^{s,a}$ in total variation distance, i.e.,* $\lim_{n\to\infty} \|\hat{\mathsf{P}}_n^{s,a} - \mathsf{P}_t^{s,a}\|_{TV} = 0$, *then it holds that* $\lim_{n\to\infty} \|V_{\mathcal{P}}^{\pi_n} - V_{\mathcal{P}}^*\|_{TV} = 0$.

Thus, we have a guarantee on the convergence of the value functions if the transition kernels converge in total variation, which holds for our IBE. Indeed, as shown in Theorem 1, the IBE $\hat{\mathsf{P}}_n^{s,a}$ converges to $\mathsf{P}_t^{s,a}$ in total variation. More importantly, the rate of policy convergence is bounded by the rate of convergence of the transition kernels. The convergence rate of our method is numerically verified to be faster than the vanilla IBE used in pure offline RL, demonstrating the effectiveness of our approach. The training and evaluation errors in the non-robust case can be derived as a special case of the robust setting, as described in the following corollary.

**Corollary 7.** *Let $V_{\mathsf{P}_t^{s,a}}^*$ be the optimal value function in the target domain, and let $\hat{\mathsf{P}}_n^{s,a}$ be a sequence of transition kernels with the corresponding optimal policies denoted by $\pi_n$. Then, we have*

*a)* $\|V_{\hat{\mathsf{P}}_n^{s,a}}^{\pi_n} - V_{\mathsf{P}_t^{s,a}}^*\|_{TV} \leq \max_{(s,a)} \frac{\|\hat{\mathsf{P}}_n^{s,a} - \mathsf{P}_t^{s,a}\|_{TV}}{(1-\gamma)^2}$

*b)* $\|V_{\mathsf{P}_t^{s,a}}^{\pi_n} - V_{\mathsf{P}_t^{s,a}}^*\|_{TV} \leq 2 \max_{(s,a)} \frac{\|\hat{\mathsf{P}}_n^{s,a} - \mathsf{P}_t^{s,a}\|_{TV}}{(1-\gamma)^2}$.

*c)* *If* $\lim_{n\to\infty} \|\hat{\mathsf{P}}_n^{s,a} - \mathsf{P}_t^{s,a}\|_{TV} = 0$, $\lim_{n\to\infty} \|V_{\hat{\mathsf{P}}_n^{s,a}}^{\pi_n} - V_{\mathsf{P}_t^{s,a}}^*\|_{TV} = 0$.

Part a) and b) of Corollary 7 characterize the convergence for the training and evaluation errors, respectively. Similar to the robust setting, part c) ensures the convergence of the non-robust value functions given that the transition kernels converge in total variation. The bounds for both non-robust errors are identical to those in the robust case presented in Theorem 4 and 5, where the non-robust case is a special case of the robust setting with the radius $R$ set to zero. Theorem 5 and Corollary 7 are experimentally validated in Appendix E.

**Remark 3.** *Although our work assumes that only a limited number of samples will typically be available from the target domain along with the side information, studying the asymptotic performance as $n \to \infty$ remains valuable even with limited samples. It provides theoretical guarantees that our estimates and learned policies converge, which gives confidence that the method will perform well under ideal conditions. This ensures that the approach will improve as more data becomes available. Also, asymptotic insights often form the foundation for finite-sample analysis, helping to derive practical bounds and understand performance in small-sample regimes. Our result in Theorem 1 complements our main theoretical results on bounding both the training and evaluation errors (Theorem 4 and 5). Specifically, the convergence of estimates in total variation ensures that these errors approach zero as the sample size increases, as established in Theorem 6.*

## 5 Numerical Experiments

We compare our approach with three existing methods as baselines: Fitted Q-Iteration (FQI) (Ernst et al., 2005), Importance Weighted Fitted Q-Iteration (IWFQI)(Tirinzoni et al., 2018), and conventional Q-learning. In FQI and Q-learning, the policy is estimated only using samples from the target domain, while IWFQI

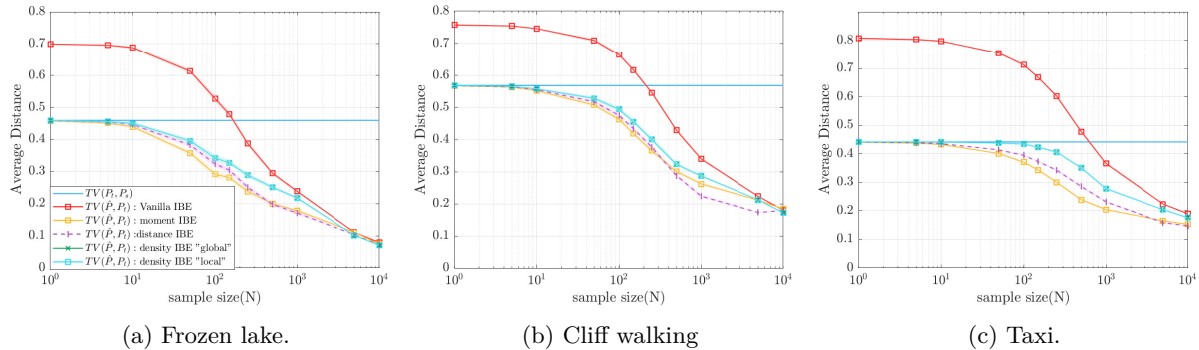

(a) Frozen lake.        (b) Cliff walking        (c) Taxi.

Figure 2: Average total variation distance between the estimated and the target domain transition kernels over 20 runs as a function of sample size, and the corresponding 95% confidence interval for the toy text environments.

leverages data from both the source and target domains, using weights (density ratios) to select the most important source domain samples. In the original work, Gaussian Processes (GPs) were utilized in IWFQI to estimate these weights, limiting applicability to continuous state spaces. Here, instead of relying on GPs, we directly use the true weights. Using the true weights for sample selection represents the optimal performance achievable with this method.

We estimate the transition kernels using the IBE programs listed in Table 1, by incorporating the prior information as the constraints for the IB methods as follows: distance IBE ($d = \|\mathsf{P}_{\mathsf{s}}^{s,a} - \mathsf{P}_{\mathsf{t}}^{s,a}\|_1$), moment IBE ($\delta = |\mu_{\mathsf{P}_{\mathsf{s}}^{s,a}}(t) - \mu_{\mathsf{P}_{\mathsf{s}}^{s,a}}(s)|$), density IBE 'global' ($B = \max(\mathsf{P}_{\mathsf{t}}^{s,a}/\mathsf{P}_{\mathsf{s}}^{s,a})$), and density IBE 'local' ($B = \mathsf{P}_{\mathsf{t}}^{s,a}/\mathsf{P}_{\mathsf{s}}^{s,a} + 1$). For our approach, we first estimate the transition kernel using the IBE, and then obtain the optimal (robust/non-robust) policy using (robust/non-robust) RL algorithms. We then evaluate the performance of the learned policies in the target domain, under both non-robust and robust settings (see Appendix C). Specifically, we estimate the non-robust value functions under the non-robust setting using the value iteration algorithm, and estimate the worst-case performance under the robust setting.

Our approach is tested on three popular toy text environments, as well as classic control problems from OpenAI Gym (Brockman et al., 2016). The toy text environments are Frozen Lake, Cliff Walking, and Taxi, while the classic control problems are Acrobot, Cart Pole, and Pendulum. A source and target domain are created that differ only in their transition kernels. See Appendix D for a detailed description of the environments used. For all experiments, the same sampling procedure is followed. We uniformly sample $N$ state-action $(s,a)$ pairs, then identify the number of times each $(s,a)$ pair was sampled. For each repetition of $(s,a)$, we sample the next state according to the corresponding target transition kernel, with different sample sizes, i.e. $\{1, 5, 10, 50, 100, 150, \dots, 10^4\}$. These samples are used to estimate the transition kernels. We initialize the estimated transition kernels with the source domain kernels, except in the vanilla IBE case, where the kernel is initialized with a uniform distribution to evaluate policy performance based solely on the given samples. The results are averaged over 20 runs.

## 5.1 Convergence of the estimators

We experimentally verify the convergence of our estimators to the target domain transition kernels, as theoretically established in Theorem 1. Figure 2 shows the average distance over all $(s,a)$ pairs between our IBE estimates $\hat{\mathsf{P}}_n^{s,a}$ and the true target transition kernels $\mathsf{P}_{\mathsf{t}}^{s,a}$, i.e., $\frac{1}{|\mathcal{S}||\mathcal{A}|} \sum_{(s,a)} TV(\hat{\mathsf{P}}_n^{s,a}, \mathsf{P}_{\mathsf{t}}^{s,a})$, averaged over 20 runs, versus the sample size for different toy text environments, including Frozen Lake, Cliff Walking, and Taxi. We also report the average distance between the source and target domain transition kernels. The results indicate that the IBE converges asymptotically to the target domain kernels as $N \to \infty$.

The results demonstrate that our estimators can converge to the true target transition kernels under different types of information. More importantly, our information-based estimators converge to the true kernel much faster compared to the baselines. This indicates that our approach is more data-efficient for transfer learning, validating our claims and theoretical results.

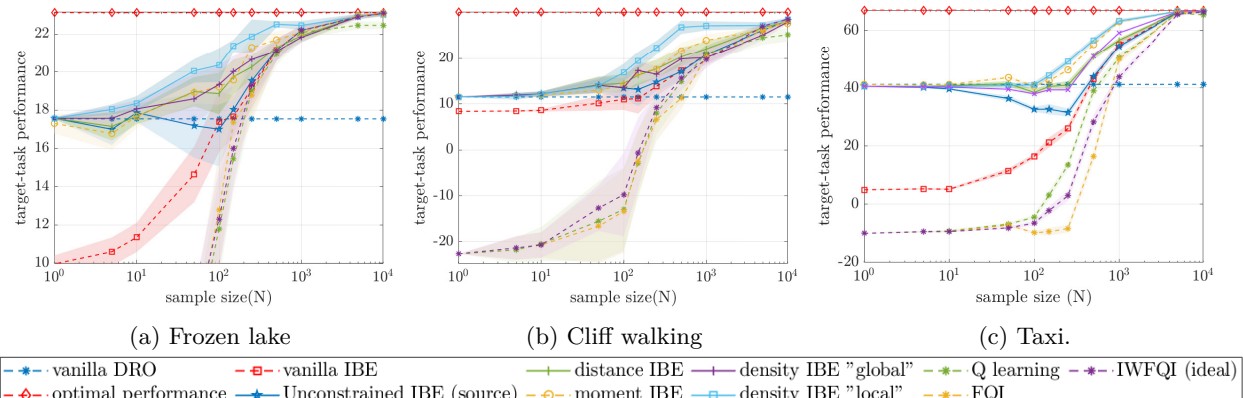

Figure 3: Target domain performance for the non-robust setting averaged over 20 runs as a function of sample size for the toy text environments, and the corresponding 95% confidence intervals.

## 5.2 Non-robust setting

In the non-robust setting, where a transition kernel is fixed, our objective is to learn a policy that maximizes the expected reward in the target domain. Since the target domain transition kernels are not directly available, we estimate them using the IB methods outlined in Table 1. We then evaluate the optimal policies learned using the estimators in the target domain. Figure 3 shows the target task performance, namely, the average value function, i.e., $\frac{1}{|\mathcal{S}|} \sum_{s \in \mathcal{S}} V^\pi(s)$, as a function of the sample size $N$ averaged over 20 runs for the toy text environments, and the corresponding 95% confidence interval. The results for the classic control problems are deferred to Appendix E.

To demonstrate the benefit of incorporating similarity side information between the two domains on the learned policy beyond initialization, we also compare it to a policy derived by initializing the unconstrained estimator with the source transition kernel. The results show that prior information can significantly enhance performance in the target domain. Furthermore, the advantage of using prior information to achieve better performance in the target domain, compared to the vanilla IBE which relies solely on the samples, is evident.

The results indicate that when prior information is incorporated, our approach not only achieves more accurate estimation but also attains higher performance in the target domain compared to baselines, including pure DRO, vanilla IBE, and other transfer learning algorithms. Specifically, the vanilla DRO approach is too conservative when the source and target domains are distinct, as the uncertainty set is excessively large. Although the vanilla IBE approach converges to the optimal performance, it requires more samples than our approach and is less effective. Among our methods, we observe that Density IBE 'local' has the best performance. We attribute its superiority to the quality of the side information regarding the relationship between the transition kernels, such as the density ratio. The more accurately we understand the relationship between both environments, the better the performance will be. Similar performance has observed for the classic control problems (see Appendix E).

## 5.3 Robust setting

In the robust setting, we account for uncertainty in the transition kernels by using an uncertainty set of distributions. The objective is to find a robust policy that optimizes the worst-case value function in the target domain, as stated in (4). We use Algorithm 1 (see Appendix C) to estimate the policy for our proposed methods, including the source and target domains with a radius $R = 0.1$. To evaluate the learned policy, we use an uncertainty set with the same radius in the target domain environment using Algorithm 2. Figure 4 presents the target task performance, i.e., $\frac{1}{|\mathcal{S}|} \sum_{s \in \mathcal{S}} V^\pi(s)$, evaluated on the target domain uncertainty set as a function of the sample size for the toy text environments. We report the average over 20 runs and the corresponding 95% confidence intervals. The results for the classic control problems are deferred to Appendix E.

As observed, our approaches outperform state-of-the-art methods. Similar to the non-robust case, the advantage of incorporating prior information is clearly reflected in the target task performance. Additionally, we compare our approaches with an over-conservative approach, where the radius of the uncertainty set centered at the source transition kernels is enlarged to include the target domain ones. The radius is set to $R + \|\mathsf{P}_{\mathsf{s}}^{s,a} - P_{\mathsf{t}}^{s,a}\|_{TV}$. As expected, the over-conservative policy performs poorly in the target domain. Similar to the non-robust case, the density IBE 'local' achieves the best performance among the estimation methods. This is attributed to the strong side information $B$, which effectively captures the relationship between the two kernels. In addition, the same conclusion can be drawn for the classic control problems, as shown in Appendix E.

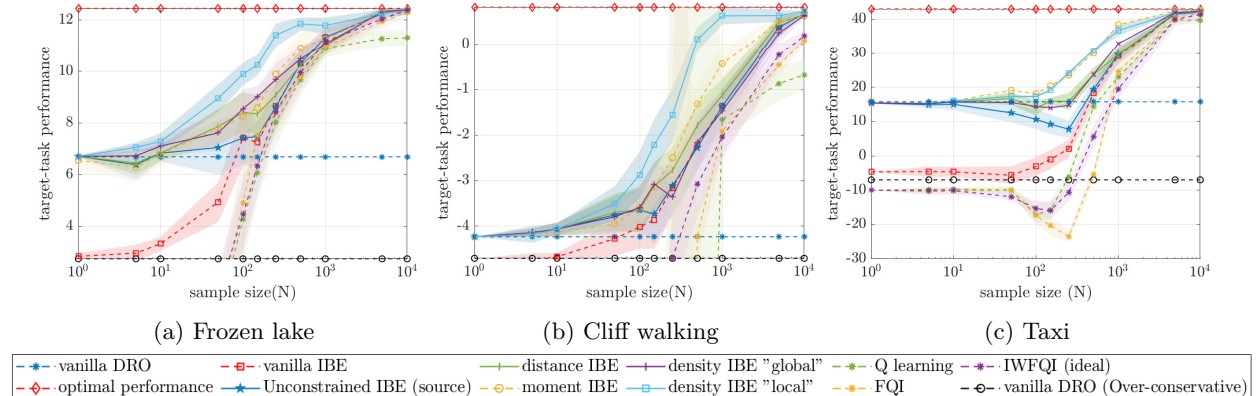

|                | (a) Frozen lake | (b) Cliff walking | (c) Taxi |
|---|---|---|---|

- - ✱ - · vanilla DRO      - - ▢ - · vanilla IBE     ─┼─ distance IBE     ─┼─ density IBE "global"     - - ✳ - Q learning     - - ✱ - · IWFQI (ideal)
- - ◇ - · optimal performance     ─★─ Unconstrained IBE (source)     - - ◇ - moment IBE     ─▢─ density IBE "local"     - - ◇ - · FQI     - - ○ - · vanilla DRO (Over-conservative)

Figure 4: Target domain performance of the robust MDPs averaged over 20 runs as a function of a sample size for the toy text environments, and the corresponding confidence intervals.

## 6 Conclusion and Future work

In this paper, we proposed a model-based approach for robust transfer learning. We considered two distinct environments, namely source and target domains, represented by two MDPs. The source domain MDP is fully accessible to the agent, while only i.i.d. samples are available from the target domain environment. Unlike previous pure RL-based or robust RL algorithms, we utilized both data from the target environment and prior side information on the relationship between the two environments. We incorporated the side information as constraints that the estimated kernels should satisfy, and estimated them through an Information-Based (IB) framework. Our framework was shown to achieve accurate estimation of the target environment with fewer samples and a tighter Cramér–Rao bound, demonstrating the efficiency and effectiveness of our approach. We further studied the impact of our estimation on the performance in the target environment, showing that our framework results in a more robust policy with better performance compared to pure offline RL and robust RL approaches. The theoretical and numerical results together imply the effectiveness of our approach in the context of transfer learning.

Several future research directions can be explored to extend this work. One promising direction is studying the rate of convergence of the IBE estimators to better understand how quickly the estimates approach the true target transition kernels as the sample size increases. Another important avenue is deriving finite sample complexity guarantees for the learned policies, which would provide theoretical bounds on performance in scenarios with limited data and further validate the robustness of the approach. Additionally, alternative definitions of the uncertainty set could be considered using metrics other than Total Variation distance, such as Chi-Square and Wasserstein distances. These metrics may offer new perspectives on the tradeoffs between robustness and performance and broaden the applicability of the method to diverse problem settings.

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

Table 2: Notation

| Notation | Description |
|---|---|
| $\mathsf{P}_{s}^{s,a}(s')$ | The source domain $(s,a)$ transition kernel at state $s'$. |
| $\mathsf{P}_{t}^{s,a}(s')$ | The target domain $(s,a)$ transition kernel at state $s'$. |
| $\|\mathsf{F} - \mathsf{Q}\|_{TV}$ | The total variational distance between the probability distributions $\mathsf{F}$ and $\mathsf{Q}$. |
| $\hat{\mathsf{P}}_{n}^{s,a}$ | The estimate of $\mathsf{P}_{t}^{s,a}$. |
| $V_{\mathcal{P}}^{\pi}$ | The worst-case value function under the policy $\pi$ over the uncertainty set $\mathcal{P}$. |
| $\mathcal{P}_{s}^{a}$ | Uncertainty set consisting of all distributions within a specified distance from the center kernel. |
| $\mathcal{P}$ | Rectangular uncertainty set of cross-product over all $(s,a)$ pairs, i.e., $\mathcal{P} = \bigotimes_{s,a} \mathcal{P}_{s}^{a}$ |

## A    Notations

Table 2 summarizes the main notation used in this work.

## B    Proofs

### B.1    Proof of Theorem 1

We first establish the following lemma on the convergence of the vanilla IBE in total variation.

**Lemma 8.** *Let $\hat{\mathsf{P}}_{n}^{s,a}$ be the vanilla IBE of $\mathsf{P}_{t}^{s,a}$, then*

$$\lim_{n\to\infty} \|\hat{\mathsf{P}}_{n}^{s,a} - \mathsf{P}_{t}^{s,a}\|_{TV} = 0. \tag{12}$$

*Proof.* By the consistency of the MLE, for a sequence $x^n = (x_1, \ldots, x_n)$ drawn i.i.d. from the discrete distribution $\mathsf{P}_{t}^{s,a}$, the MLE estimate of $\mathsf{P}_{t}^{s,a}$ is given by

$$\hat{\mathsf{P}}_{j,n}^{s,a} = \frac{n_j}{n}, \quad j = 1, \ldots, k \tag{13}$$

where $n_j := \sum_{i=1}^{n} \mathbf{1}(x_i = j)$ represents the number of times state $j$ is observed in the sample.

By the Strong Law of Large Numbers (SLLN), we have that

$$\frac{n_j}{n} = \frac{1}{n}\sum_{i=1}^{n} \mathbf{1}(x_i = j) \xrightarrow{\text{a.s.}} \mathsf{P}_{t,j}^{s,a}, \quad \forall j \in [k].$$

By the almost sure convergence, $\Pr(w \in \Omega : \lim_{n\to\infty} \frac{n_j}{n} = \mathsf{P}_{t,j}^{s,a}) = 1$, where $\Omega$ is the sample space. Hence, for any $\epsilon > 0$, $\exists N_j$ such that

$$\left|\frac{n_j}{n} - \mathsf{P}_{t,j}^{s,a}\right| \leq \frac{\epsilon}{k}, \quad \forall n \geq N_j. \tag{14}$$

Let $N = \max(N_1, \ldots, N_k)$. It follows that

$$\sum_{j=1}^{k} \left|\frac{n_j}{n} - \mathsf{P}_{t,j}^{s,a}\right| \leq \epsilon, \quad \forall n \geq N. \tag{15}$$

By equation 13, $\sum_{j=1}^{k} \left|\hat{\mathsf{P}}_{j,n}^{s,a} - \mathsf{P}_{t,j}^{s,a}\right| \leq \epsilon, \quad \forall n \geq N$. Hence, $\|\hat{\mathsf{P}}_{n}^{s,a} - \mathsf{P}_{t}^{s,a}\|_{TV} \to 0$, which completes the proof. $\square$

Now, we are ready to prove Theorem 1.

*Proof.* : We prove the results for each IBE method listed in Table 1. Both the vanilla IBE and the IBE programs have a unique solution for each $n$, as the objective is strictly concave and the constraints are convex. We use the results of Theorem 8 which states that, for any $\varepsilon > 0$, $\exists N_0(\varepsilon)$ such that $\|\hat{\mathsf{P}}_n^{s,a} - \mathsf{P}_t^{s,a}\|_{TV} \leq \varepsilon$, $\forall n > N_0$, where $\hat{\mathsf{P}}_n^{s,a}$ is an optimal vanilla IBE, to prove each IBE case.

**Distance IBE.**

Let $d_0^{s,a} = \|\mathsf{P}_s^{s,a} - \mathsf{P}_t^{s,a}\|_1 = 2\|\mathsf{P}_s^{s,a} - \mathsf{P}_t^{s,a}\|_{TV}$ . Since $\mathsf{P}_t^{s,a}$ is an interior point of the constraint set, we can choose $\varepsilon \leq d - \frac{d_0^{s,a}}{2}$. Hence, for sufficiently large $n$,

$$\|\hat{\mathsf{P}}_n^{s,a} - \mathsf{P}_s^{s,a}\|_{TV} \overset{(a)}{\leq} \|\hat{\mathsf{P}}_n^{s,a} - \mathsf{P}_t^{s,a}\|_{TV} + \|\mathsf{P}_s^{s,a} - \mathsf{P}_t^{s,a}\|_{TV} \overset{(b)}{\leq} \varepsilon + \frac{d_0^{s,a}}{2} \overset{(c)}{\leq} d \tag{16}$$

where (a) follows from the triangular inequality, (b) by Theorem 8, and (c) from the choice of $\varepsilon$. Hence, $\hat{\mathsf{P}}_n^{s,a}$ satisfies the distance constraint. Therefore, by the uniqueness of $(\hat{\mathsf{P}}_n^{s,a})^*$ (the solution to (5)), it follows that $\hat{\mathsf{P}}_n^{s,a}$ solves the distance IBE program. Consequently, $(\hat{\mathsf{P}}_n^{s,a})^* \to \mathsf{P}_t^{s,a}$ in total variation.

**Density IBE.**
We first denote the density ratio between $\mathsf{P}_s^{s,a}$ and $\mathsf{P}_t^{s,a}$ by $B_0^{s,a}$, i.e.

$$B_0^{s,a} = \frac{\mathsf{P}_t^{s,a}}{\mathsf{P}_s^{s,a}} \tag{17}$$

By Theorem 8, each entry of $\hat{\mathsf{P}}_n^{s,a}$ converges to the corresponding entry of $\mathsf{P}_t^{s,a}$. For sufficiently large $n$, we have

$$\left| \frac{\hat{\mathsf{P}}_{n,i}^{s,a}}{\mathsf{P}_{t,i}^{s,a}} - 1 \right| \leq \varepsilon \implies \frac{\hat{\mathsf{P}}_{n,i}^{s,a}}{\mathsf{P}_{t,i}^{s,a}} \leq 1 + \varepsilon \tag{18}$$

where $\hat{\mathsf{P}}_{n,i}^{s,a}$ and $\mathsf{P}_{t,i}^{s,a}$ are the $i^{th}$ element of $\hat{\mathsf{P}}_n^{s,a}$ and $\mathsf{P}_t^{s,a}$, respectively. Then, we get that,

$$\frac{\hat{\mathsf{P}}_{n,i}^{s,a}}{\mathsf{P}_{s,i}^{s,a}} \overset{(a)}{\leq} B_{0,i}^{s,a}(1+\varepsilon) \overset{(b)}{\leq} B_i \tag{19}$$

where $B_{0,i}^{s,a}$, $B_i$ and $\mathsf{P}_{s,i}^{s,a}$ are the $i^{th}$ entry of $B_0^{s,a}$, $B$, and $P_s^{s,a}$, respectively. Here: (a) follows from (17) and (18), while (b) by the choice of $\varepsilon$. We conclude that $\hat{\mathsf{P}}_n^{s,a}$ satisfies the Density IBE's constraint. Since $(\hat{\mathsf{P}}_n^{s,a})^*$ is a unique solution of (5), $\hat{\mathsf{P}}_n^{s,a}$ solves the density IBE. Thus, the result follows.

**Moment IBE.**
We define $\delta_0^{s,a}$ as the absolute difference between the embedding means of $\mathsf{P}_s^{s,a}$ and $\mathsf{P}_t^{s,a}$, i.e.

$$\delta_0^{s,a} = \left| \mu_{\mathsf{P}_t^{s,a}} - \mu_{\mathsf{P}_s^{s,a}} \right| . \tag{20}$$

Now, we prove that each entry of $|\mu_{\hat{\mathsf{P}}_n^{s,a}} - \mu_{\mathsf{P}_t^{s,a}}|$ goes to zero as $n \to \infty$. Suppose $\phi$ maps the state space (of $K$ states) to a high-dimensional feature space of dimension $M$, and consider the matrix $A$ of features of size $M \times K$, i.e.

$$A = \begin{bmatrix} \mathbf{a}_1 \\ \vdots \\ \mathbf{a}_M \end{bmatrix} = \begin{bmatrix} a_{1,1} & \cdots & a_{1,K} \\ \vdots & \ddots & \vdots \\ a_{M,1} & \cdots & a_{M,K} \end{bmatrix} \tag{21}$$

where $\phi(x_i) = (a_{1,i}, \ldots, a_{M,i})^\top, x_i \in \mathcal{S}, i \leq K$.

$$|\mu_{\hat{\mathsf{P}}_n^{s,a}} - \mu_{\mathsf{P}_t^{s,a}}| = |A(\hat{\mathsf{P}}_n^{s,a} - \mathsf{P}_t^{s,a})|$$

$$= \begin{bmatrix} |\sum_{i=1}^K a_{1,i}(\hat{\mathsf{P}}_{n,i}^{s,a} - \mathsf{P}_{t,i}^{s,a})| \\ \vdots \\ |\sum_{i=1}^K a_{M,i}(\hat{\mathsf{P}}_{n,i}^{s,a} - \mathsf{P}_{t,i}^{s,a})| \end{bmatrix}$$

$$\leq \begin{bmatrix} \|\mathbf{a}_1^\top(\hat{\mathsf{P}}_n^{s,a} - \mathsf{P}_t^{s,a})\|_1 \\ \vdots \\ \|\mathbf{a}_M^\top(\hat{\mathsf{P}}_n^{s,a} - \mathsf{P}_t^{s,a})\|_1 \end{bmatrix}$$

$$\overset{(*)}{\leq} \begin{bmatrix} \|A\|_\infty\|(\hat{\mathsf{P}}_n^{s,a} - \mathsf{P}_t^{s,a})\|_1 \\ \vdots \\ \|A\|_\infty\|(\hat{\mathsf{P}}_n^{s,a} - \mathsf{P}_t^{s,a})\|_1 \end{bmatrix}$$

$$= 2\|A\|_\infty\|\hat{\mathsf{P}}_n^{s,a} - \mathsf{P}_t^{s,a}\|_{TV}\mathbf{1}, \tag{22}$$

where $\mathbf{1}$ is the vector of all ones and (*) follows from Hölder's inequality. Hence, by Theorem 8, we have:

$$\lim_{n\to\infty} |\mu_{\hat{\mathsf{P}}_n^{s,a}} - \mu_{\mathsf{P}_t^{s,a}}| = 0. \tag{23}$$

For sufficiently large $n$,

$$|\mu_{\hat{\mathsf{P}}_n^{s,a}} - \mu_{\mathsf{P}_s^{s,a}}| = |A(\hat{\mathsf{P}}_n^{s,a} - \mathsf{P}_s^{s,a})|$$

$$= |A(\hat{\mathsf{P}}_n^{s,a} - \mathsf{P}_t^{s,a} + \mathsf{P}_t^{s,a} - \mathsf{P}_s^{s,a})|$$

$$\overset{(a)}{\leq} |A(\hat{\mathsf{P}}_n^{s,a} - \mathsf{P}_t^{s,a})| + |A(\mathsf{P}_t^{s,a} - \mathsf{P}_s^{s,a})|$$

$$\overset{(b)}{\leq} \varepsilon + \delta_0^{s,a}$$

$$\overset{(c)}{\leq} \delta \tag{24}$$

where (a) follows by the triangular inequality, (b) by using (20) and 23, and (c) by the choice $\varepsilon$. Then, $(\hat{\mathsf{P}}_n^{s,a})^*$ satisfies the moment constraints. The result follows by the uniqueness of $(\hat{\mathsf{P}}_n^{s,a})^*$.

## B.2   Proof of Lemma 2

*Proof.* : To simplify notation, we denote $\mathsf{P}_t^{s,a}$ by $\mathsf{P}'$. Let $S$ be a random vector that takes one of the values from $\{e_1, \ldots, e_k\}$, where $e_j, j = 1, \ldots, k$, is a vector with a 1 in the $j$-th position and 0 elsewhere (one-hot encoding). The entries are denoted $s_j$, where $\Pr(s_j = 1) = \Pr(S = e_j) = \mathsf{P}'_j$, so the vector $\mathsf{P}' = [\mathsf{P}'_1, \ldots, \mathsf{P}'_k]^T$, and $\sum_{j=1}^k \mathsf{P}'_j = 1$. To compute the Fisher Information Matrix (FIM), we define the likelihood

$$\Pr(S|\mathsf{P}') = \prod_{j=1}^k \mathsf{P}'^{s_j}_j.$$

Hence,

$$\log \Pr(s|\mathsf{P}') = \sum_{j=1}^k s_j \log \mathsf{P}'_j.$$

Therefore, the FIM, $I_S(\mathsf{P}') = \mathbb{E}[\nabla \log \Pr(s|\mathsf{P}')\nabla \log \Pr(s|\mathsf{P}')^T]$, is given by:

$$I_S(\mathsf{P}') = \mathbb{E}\left(\begin{bmatrix} s_1/\mathsf{P}'_1 \\ \vdots \\ s_k/\mathsf{P}'_k \end{bmatrix} \begin{bmatrix} s_1/\mathsf{P}'_1 & \cdots & s_k/\mathsf{P}'_k \end{bmatrix}\right).$$

This simplifies to

$$I_S(\mathsf{P}') = \mathbb{E} \begin{bmatrix} (s_1/\mathsf{P}'_1)^2 & s_1 s_2/(\mathsf{P}'_1\mathsf{P}'_2) & \cdots & s_1 s_k/(\mathsf{P}'_1\mathsf{P}'_k) \\ \vdots & \ddots & \ddots & \vdots \\ s_k s_1/(\mathsf{P}'_k\mathsf{P}'_1) & \cdots & \cdots & (s_k/\mathsf{P}'_k)^2 \end{bmatrix}.$$

Now, $\mathbb{E}[(s_i/\mathsf{P}'_i)^2] = \frac{1}{\mathsf{P}'^2_i}\mathbb{E}[s_i^2] = 1/\mathsf{P}'^2_i(1\cdot\mathsf{P}'_i + 0) = 1/\mathsf{P}'_i, i = 1,\ldots,k$ and $\mathbb{E}[(s_i s_j/\mathsf{P}'_i\mathsf{P}'_j)] = (1/\mathsf{P}'_i\mathsf{P}'_j)\mathbb{E}[s_i s_j] = 0$, for $i \neq j$. Thus,

$$I_S(\mathsf{P}') = \mathrm{diag}\left(\frac{1}{\mathsf{P}'_1},\ldots,\frac{1}{\mathsf{P}'_k}\right).$$

If we observe $n$ i.i.d. vectors $S$, we get that $J(\mathsf{P}') = nI_S(\mathsf{P}')$.

### B.3 Proof of Theorem 3

We use the same notation simplifications as in the proof of Lemma 2. We start by bringing in the following lemma.

**Lemma 9** (Theroem 2, (Marzetta, 1993)). *Let the $k$-parameter vector $\hat{\theta}$ be a constrained unbiased estimator of the parameter $\theta$, with $M$ constrains $f(\theta) = 0$ for some real valued function $f$. Then, the CRB is given by:*

$$C(\theta) = J^{-1} - J^{-1}F(F^T J^{-1}F)^{-1}F^T J^{-1}. \tag{25}$$

*where $J$ is the FIM and $F$ is $k \times M$ gradient matrix w.r.t $\theta$.*

*Proof.* :

**Regular probability constraint** $(\sum_i^k \mathsf{P}'_i = 1)$:

Due to the normalization constraint $\sum_i^k \mathsf{P}'_i = 1$, the probabilities $\mathsf{P}'_i$ are dependent. To handle this, we express $\mathsf{P}'_k = 1 - \sum_i^{k-1}\mathsf{P}'_i$, and use a reduced parameter vector $\theta = [\mathsf{P}'_1,\ldots,\mathsf{P}'_{k-1}]$. Computing the FIM, we get:

$$J = n\left(\mathrm{diag}\left(\left[\frac{1}{\mathsf{P}'_1},\ldots,\frac{1}{\mathsf{P}'_{k-1}}\right]\right) + \frac{1}{\mathsf{P}'_k}\mathbf{1}_{k-1}\mathbf{1}_{k-1}^\top\right) \tag{26}$$

where $\mathbf{1}_{k-1}$ is a vector of all ones of size $k-1$. Using Sherman-Morrison-Woodbury formula (Petersen & Pedersen, 2008), we compute the the CRB matrix, i.e., $J^{-1}$:

$$C(\mathsf{P}') = J^{-1} = \frac{1}{n}\left(\mathrm{diag}(\mathsf{P}') - \mathsf{P}'(\mathsf{P}')^\top\right). \tag{27}$$

**Moment constraint** $(\mathbb{E}_{\mathsf{P}'}[\phi(x)] = \mu$, where $x \in \mathcal{S}$ and $\phi : \mathcal{S} \to \mathbb{R}^m$, and $\sum_i^k \mathsf{P}'_i = 1)$: By Lemma 9, the CRB matrix $C$ is given by:

$$C(\mathsf{P}') = C_R^{-1} - C_R^{-1}F(F^T C_R^{-1}F)^{-1}F^\top C_R^{-1}. \tag{28}$$

Here, we use $C_R$ to denote the CRB in (27), incorporating only the regular probability constraint. The constraints, $f(\mathsf{P}') = \mathbb{E}_{\mathsf{P}'}[\phi(x)] - \mu = 0 \in \mathbb{R}^m$, can be rewritten as:

$$\begin{aligned} f(\mathsf{P}') &= \sum_{i=1}^{k-1}\phi(x_i)\mathsf{P}'_i + \phi(x_k)\mathsf{P}'_k - \mu = 0 \\ &\overset{(*)}{=} \sum_{i=1}^{k-1}(\phi(x_i) - \phi(x_k))\mathsf{P}'_i + \phi(x_k) - \mu = 0 \\ &= \sum_{i=1}^{k-1}a_i\mathsf{P}'_i + b = 0 \quad (m \text{ constraints}) \end{aligned} \tag{29}$$

where (*) follows from the constraint $\mathsf{P}'_k = 1 - \sum_{i=1}^{k-1} \mathsf{P}'_i$, $a_i = \phi(x_i) - \phi(x_k) \in \mathbb{R}^m$, and $b = \phi(x_k) - \mu \in \mathbb{R}^m$.

Let the constraint set $f(\mathsf{P}') = [f_1, \ldots, f_m]$, then the gradient matrix F of size $m \times (k-1)$ is given by:

$$
\begin{bmatrix}
\frac{df_1}{d\mathsf{P}'_1} & \cdots & \frac{df_1}{d\mathsf{P}'_{k-1}} \\
\vdots & \ddots & \vdots \\
\frac{df_m}{d\mathsf{P}'_1} & \cdots & \frac{df_m}{d\mathsf{P}'_{k-1}}
\end{bmatrix}
=
\begin{bmatrix}
a_{1,1} & \cdots & a_{1,k-1} \\
\vdots & \ddots & \vdots \\
a_{m,1} & \cdots & a_{m,k-1}
\end{bmatrix}
\tag{30}
$$

where $a_i = [a_{1,i}, \ldots, a_{m,i}]$. Then,

$$
\mathbb{S} \triangleq F^T C_R^{-1} F = \frac{1}{n} \left( \sum_{i=1}^{k-1} a_i a_i^\top \mathsf{P}'_i - \left( \sum_{i=1}^{k-1} a_i \mathsf{P}'_i \right) \left( \sum_{j=1}^{k-1} a_j \mathsf{P}'_j \right) \right)
$$

$$
= \frac{1}{n} \operatorname{Cov}(a)
\tag{31}
$$

where $a$ is a vector that takes the value $a_i$ with probability $\mathsf{P}'_i$, and Cov(a) is its covariance matrix. Also, $C_R^{-1} F = \frac{1}{n}(\operatorname{diag}(\mathsf{P}'_i) - \mathsf{P}'(\mathsf{P}')^\top)F = \frac{1}{n}(\mathbb{P} - \mathsf{P}'\bar{a}^\top)$, where $\mathbb{P}$ is the $(k-1) \times m$ matrix with rows $\mathsf{P}'_i a_i$ and $\bar{a} = \sum_{i=1}^k a_i \mathsf{P}^{s,a}_{t,i}$. Thus,

$$
C(\mathsf{P}') = C_R^{-1} - (C_R^{-1}F)\mathbb{S}^{-1}(C_R^{-1}F)^\top
$$

and

$$
\operatorname{Var}[\mathsf{P}'_i] \geq (C_R^{-1})_{ii} - (C_R^{-1}F)_i^\top \mathbb{S}^{-1}(C_R^{-1}F)_i
$$

where $(C_R^{-1})_{ii} = \frac{\mathsf{P}'_i(1-\mathsf{P}'_i)}{n}$ and $(C_R^{-1}F)_i^\top = \frac{\mathsf{P}'_i}{n}(a_i - \bar{a})$ is the $i^{th}$ row of $(C_R^{-1}F)$. Then,

$$
\Delta_i := (C_R^{-1}F)_i^\top \mathbb{S}^{-1}(C_R^{-1}F)_i
$$

$$
= \left( \frac{\mathsf{P}'_i}{n}(a_i - \bar{a}) \right)^\top \mathbb{S}^{-1} \left( \frac{\mathsf{P}'_i}{n}(a_i - \bar{a}) \right)
$$

$$
= \frac{(\mathsf{P}'_i)^2}{n^2}(a_i - \bar{a})^\top \mathbb{S}^{-1}(a_i - \bar{a}).
\tag{32}
$$

Since $\mathbb{S}$ is positive definite (assuming Cov(a) is full rank), $\Delta_i \geq 0$, indicating that the lower bound on the variance of the estimator is reduced with the incorporation of the moment constraint.

We remark that, if Cov(a) is not a full rank matrix, then $\mathbb{S}^{-1}$ can be replaced with the Moore-Penrose pseudoinverse $\mathbb{S}^\dagger$, i.e., $C(\mathsf{P}') = C_R^{-1} - (C_R^{-1}F)\mathbb{S}^\dagger(C^{R-1}F)^\top$. The matrix $\Delta := (C_R^{-1}F)\mathbb{S}^\dagger(C_R^{-1}F)^\top$ is positive semi-definite (PSD), so

$$
C(\mathsf{P}') = C_R^{-1} - \Delta \preceq C_R^{-1},
$$

so the constrained matrix is less than or equal to the unconstrained one in the PSD ordering, indicating that the reduction in the lower bound on the variance still holds.

### B.4  Proof of Theorem 4

*Proof.* Considering any state $s$, we have that

$$
|V^{\pi_n}_{\mathcal{P}_n}(s) - V^{\pi^*}_{\mathcal{P}}(s)| \leq \max_\pi |V^\pi_{\mathcal{P}_n}(s) - V^\pi_{\mathcal{P}}(s)|
\tag{33}
$$

where the inequality is from the fact that $|\max f - \max g| \leq \max|f - g|$.

For any fixed policy $\pi$,

$$
\begin{aligned}
|V_{\mathcal{P}_n}^\pi(s) - V_{\mathcal{P}}^\pi(s)| &\overset{(a)}{=} |r_\pi(s) + \gamma\sigma_{(\mathcal{P}_n)_s^{\pi(s)}}(V_{\mathcal{P}_n}^\pi) - r_\pi(s) - \gamma\sigma_{\mathcal{P}_s^{\pi(s)}}(V_{\mathcal{P}}^\pi)| \\
&= |\gamma\sigma_{(\mathcal{P}_n)_s^{\pi(s)}}(V_{\mathcal{P}_n}^\pi) - \gamma\sigma_{\mathcal{P}_s^{\pi(s)}}(V_{\mathcal{P}}^\pi)| \\
&= |\gamma\sigma_{(\mathcal{P}_n)_s^{\pi(s)}}(V_{\mathcal{P}_n}^\pi) - \gamma\sigma_{\mathcal{P}_s^{\pi(s)}}(V_{\mathcal{P}_n}^\pi) + \gamma\sigma_{\mathcal{P}_s^{\pi(s)}}(V_{\mathcal{P}_n}^\pi) - \gamma\sigma_{\mathcal{P}_s^{\pi(s)}}(V_{\mathcal{P}}^\pi)| \\
&\overset{(b)}{\leq} \gamma\|V_{\mathcal{P}_n}^\pi - V_{\mathcal{P}}^\pi\| + \gamma|\sigma_{(\mathcal{P}_n)_s^{\pi(s)}}(V_{\mathcal{P}_n}^\pi) - \sigma_{\mathcal{P}_s^{\pi(s)}}(V_{\mathcal{P}_n}^\pi)|,
\end{aligned}
\tag{34}
$$

where $(a)$ is from the robust Bellman equation, and $(b)$ is from the Lipschitz smoothness of $\sigma_{\mathcal{P}}(\cdot)$ (Wang & Zou, 2021).

Consider the second term $\|\sigma_{(\mathcal{P}_n)_s^{\pi(s)}}(V_{\mathcal{P}_n}^\pi) - \sigma_{\mathcal{P}_s^{\pi(s)}}(V_{\mathcal{P}_n}^\pi)\|$. Note that the support function can be solved by its dual form:

$$
\sigma_{\mathcal{P}}(V) = \min_{\mathsf{P}\in\mathcal{P}}\mathsf{P}^\top V = \max_{\alpha\geq 0}\{\mathsf{P}_0^\top(V-\alpha) - R\mathbf{Span}(V-\alpha)\},
\tag{35}
$$

where $\mathsf{P}_0$ is the center of $\mathcal{P}$ and $R$ is the radius. Hence, (34 can be further bounded as

$$
|V_{\mathcal{P}_n}^{\pi_n}(s) - V_{\mathcal{P}}^{\pi^*}(s)| \leq \gamma\|V_{\mathcal{P}_n}^\pi - V_{\mathcal{P}}^\pi\| + \gamma|(\hat{\mathsf{P}}_n^{s,\pi(s)} - \mathsf{P}_t^{s,\pi(s)})^\top V_{\mathcal{P}_n}^\pi|,
\tag{36}
$$

which is from the fact that $|\max_x f(x) - \max_y g(y)| \leq \max_x |f(x) - g(x)|$. Note that (36) holds for any $s$ and any policy $\pi$, thus

$$
\|V_{\mathcal{P}_n}^{\pi_n} - V_{\mathcal{P}}^{\pi^*}\| \leq \gamma\|V_{\mathcal{P}_n}^\pi - V_{\mathcal{P}}^\pi\| + \gamma\max_{(s,a)}\|(\hat{\mathsf{P}}_n^{s,a} - \mathsf{P}_t^{s,a})^\top V_{\mathcal{P}_n}^\pi\|.
\tag{37}
$$

Then, recursively applying (37) implies that

$$
\|V_{\mathcal{P}_n}^{\pi_n} - V_{\mathcal{P}}^{\pi^*}\| \leq \sum_{t=1}^\infty \frac{\gamma^t}{1-\gamma}\max_{(s,a)}\|\hat{\mathsf{P}}_n^{s,a} - \mathsf{P}_t^{s,a}\| = \max_{(s,a)}\frac{\|\hat{\mathsf{P}}_n^{s,a} - \mathsf{P}_t^{s,a}\|}{(1-\gamma)^2}.
\tag{38}
$$

$\square$

## B.5 Proof of Theorem 5

*Proof.* For each transition kernel $\hat{\mathsf{P}}_n^{s,a}$ in the sequence, denote the optimal policy w.r.t. it by $\pi_n$, i.e.,

$$
\pi_n = \arg\max_\pi V_{\mathcal{P}_n}^\pi.
\tag{39}
$$

Then, we have that

$$
\|V_{\mathcal{P}}^{\pi_n} - V_{\mathcal{P}}^{\pi^*}\| \leq \|V_{\mathcal{P}}^{\pi_n} - V_{\mathcal{P}_n}^{\pi_n}\| + \|V_{\mathcal{P}_n}^{\pi_n} - V_{\mathcal{P}}^{\pi^*}\|
\tag{40}
$$

Following the same technique in the proof as of Theorem 4, we know that

$$
\|V_{\mathcal{P}_n}^{\pi_n} - V_{\mathcal{P}}^{\pi^*}\| \leq \max_\pi\|V_{\mathcal{P}_n}^\pi - V_{\mathcal{P}}^\pi\| \leq \max_{(s,a)}\frac{\|\hat{\mathsf{P}}_n^{s,a} - \mathsf{P}_t^{s,a}\|}{(1-\gamma)^2}
\tag{41}
$$

and

$$
\|V_{\mathcal{P}}^{\pi_n} - V_{\mathcal{P}_n}^{\pi_n}\| \leq \max_\pi\|V_{\mathcal{P}}^\pi - V_{\mathcal{P}_n}^\pi\| \leq \sum_{t=1}^\infty \frac{\gamma^t}{1-\gamma}\max_{(s,a)}\|\hat{\mathsf{P}}_n^{s,a} - \mathsf{P}_t^{s,a}\| = \max_{(s,a)}\frac{\|\hat{\mathsf{P}}_n^{s,a} - \mathsf{P}_t^{s,a}\|}{(1-\gamma)^2}.
\tag{42}
$$

which completes the proof.

$\square$

## B.6 Proof of Theorem 6

*Proof.* : The proof follows directly from the error bounds provided in Theorem 4 and 5.

## C  Algorithms

To estimate the optimal policy of the worst-case value function as in Eq. (4), we utilize the robust value iteration algorithm presented in Algorithm 1, where $\sigma_{\mathcal{P}_s^a}(V) \triangleq \min_{\mathsf{P} \in \mathcal{P}_s^a} \mathsf{P}^\top V$ is the support function of $V$ on $\mathcal{P}_s^a$. Moreover, Algorithm 1 can also be used to solve for the optimal policy of the value function (3) if we set $\mathcal{P}_s^a = \{\mathsf{P}^{s,a}\}$, in which case $\sigma_{\mathcal{P}_s^a}(V) = \mathsf{P}^{s,a\top} V$

---
**Algorithm 1** Robust Policy Estimation

**Input**: $\gamma, V_0(s) = 0, \forall s, T$

1: **for** $t = 0, 1, ..., T - 1$ **do**
2:    **for** all $s \in \mathcal{S}$ **do**
3:       $V_{t+1}(s) \leftarrow \max_{a \in \mathcal{A}} \left\{ r(s,a) + \gamma \sigma_{\mathcal{P}_s^a}(V_t) \right\}$
4:    **end for**
5: **end for**
6: **for** $s \in \mathcal{S}$ **do**
7:    $\pi_T(s) \leftarrow \arg\max_{a \in \mathcal{A}} \left\{ r(s,a) + \gamma \sigma_{\mathcal{P}_s^a}(V_T) \right\}$
8: **end for**
9: **return** $V_T, \pi_T$

---

Based on a similar approach as in policy estimation, any policy $\pi$ can be evaluated on a general uncertainty set, as shown in Algorithm 2.

---
**Algorithm 2** Robust Policy Evaluation

**Input**: $\pi, \gamma, V_0(s) = 0, \forall s, T$

1: **for** $t = 0, 1, ..., T - 1$ **do**
2:    **for** all $s \in \mathcal{S}$ **do**
3:       $V_{t+1}(s) \leftarrow \mathbb{E}_\pi[r(s,A) + \gamma \sigma_{\mathcal{P}_s^A}(V_t)]$
4:    **end for**
5: **end for**
6: **return** $V_T$

---

## D  Description of Environments

### D.1  Toy text environments

**Frozen lake environment**: This environment is depicted in Figure 5a. We consider a $4 \times 4$ frozen lake grid, consisting of a start state, an end state, two hole states, and one prize state, where each square represents a state. The main goal is to traverse the frozen lake from the start state to the end state without falling into holes. The agent can take four actions: right, left, up and down. The agent loses '-1' if it steps on a hole, receives '+5' for stepping on the prize state, and '-0.04' for any other frozen state. Transitions to the next state occur with a certain probability upon taking an action.

**Cliff walking environment**: The goal of the agent in the cliff walking environment is to reach the end (prize) state, starting from the first state while avoiding the cliff. The environment consists of 38 states: a start state, an end state, and 36 other states. The agent incurs a penalty of '-100' if it steps on the cliff and receives '+50' upon reaching the end state. For all other states, the agent receives a reward of '-1'. A schematic of this environment is depicted in Figure 5c.

Table 3: A Description of the source and target domain transition kernels for toy-text environments.

| Transition kernel | Description |
|---|---|
| $\mathsf{P}_s^{s,a}(s_{intended})$ | The agent moves to the state in the intended direction with probability $(1-\alpha)(r_s + 2\frac{(1-r_s)}{|\mathcal{S}|})$ |
| $\mathsf{P}_s^{s,a}(s_{opposite})$ | The agent may move to the state opposite the intended direction with probability $\alpha(r_s + 2\frac{(1-r_s)}{|\mathcal{S}|})$ |
| $\mathsf{P}_s^{s,a}(s_{other})$ | The agent may move to all other states with probability $\frac{(1-r_s)}{|\mathcal{S}|}$ |
| $\mathsf{P}_t^{s,a}(s_{intended})$ | The agent moves to the state in the intended direction with probability $\alpha(r_t + 2\frac{(1-r_t)}{|\mathcal{S}|})$ |
| $\mathsf{P}_t^{s,a}(s_{opposite})$ | The agent may move to the state opposite the intended direction with probability $(1-\alpha)(r_t + 2\frac{(1-r_t)}{|\mathcal{S}|})$ |
| $\mathsf{P}_t^{s,a}(s_{other})$ | The agent may move to all other states with probability $\frac{(1-r_t)}{|\mathcal{S}|}$ |

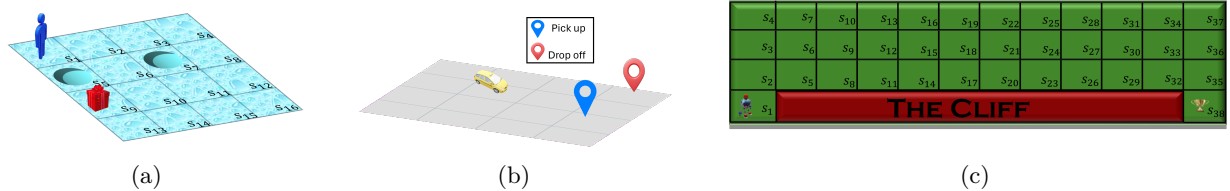

(a)  (b)  (c)

Figure 5: A depiction OpenAI Gym toy text environments:**(a)** Frozen lake environment **(b)** Taxi environment**(c)** Cliff walking environment

**Taxi environment**: In the taxi environment, the goal for the agent (taxi) is to navigate through the grid using movement actions (up, down, left, right), locate the passenger, pick up the passenger, and later drop off the passenger at the destination using pick-up and drop-off actions, respectively. In our experiment, we consider a $6 \times 6$ grid with fixed pick-up and drop-off locations, as shown in Figure 5b. The passenger can be located either at the pick-up or drop-off locations, or in the taxi, resulting in a total of 108 states and 6 possible actions. The agent is penalized '-1' for each movement action taken. Wrongful pick-up and drop-off actions incur a penalty of '-10'. The agent receives a reward of '+20' for successfully picking up the passenger from the designated location and '+50' for successfully dropping off the passenger, signifying task completion.

A detailed description of the transition kernels for both the source and target domains is provided in Table 3. We define $s_{intended}$ as the next state the agent is expected to move to when it takes action $a$ starting from state $s$. Conversely, $s_{opposite}$ refers to the state opposite to $s_{intended}$. For example, consider the frozen lake environment in the source domain. If the agent is at state $s_2$ and takes action $a = $ 'right,' it will move to $s_{intended} = s_3$ with probability $\mathsf{P}_s^{s,a}(s_{intended})$, to $s_{opposite} = s_1$ with probability $\mathsf{P}_s^{s,a}(s_{opposite})$, and to any other state $s_{other}$ with probability $\mathsf{P}_s^{s,a}(s_{other})$. For the frozen lake environment, we use $r_s = 0.3$, $r_t = 0.8$, and $\alpha = 0.7$. For cliff walking, we set $r_s = 0.8$, $r_t = 0.2$, and $\alpha = 0.3$. For the taxi environment, we choose $r_s = 0.4$, $r_t = 0.8$, and $\alpha = 0.2$. To ensure validity, we normalize each $\mathsf{P}_s^{s,a}$ and $\mathsf{P}_t^{s,a}$ so that they sum to 1.

## D.2   Classic control problems

We consider three classical control problems from OpenAIGym: Cart pole, Acrobot, and Pendulum. We have made some modifications to the environments to suit our problem setup.

**Cart Pole environment**: The goal is to balance a pole attached upright to the cart by applying forces to the left or right of the cart (Barto et al., 1983). The action space consists of two actions: right and left. The observation space is continuous and consists of a tuple of four quantities: Cart Position, Cart Velocity, Pole Angle, Pole Angular Velocity. Their corresponding ranges are $([-4.8, 4.8], (-\infty, \infty), [-24^o, 24^o], (-\infty, \infty))$. We discretize the observation space, with each combination of values representing a distinct state. The new ranges for each quantity are $([-4.8, 4.8], (-0.5, 0.5), [-24^o, 24^o], (-5, 5))$, and the number of discrete values is $(4, 4, 5, 3)$, respectively. A reward of '25' is received if the pole angle is 0 and the cart velocity is between -0.17 and +0.17. An additional reward of '25' is granted if the cart position is between -1.6 and 1.6. When the pole angle is between $(-12^o, 12^o)$ and the cart velocity is between -0.17 and +0.17, a reward of '10' is received. For all other cases, no rewards are received. The Cart Pole environment is depicted in Figure 6a.

Table 4: A Description of the source and target domain transition kernels for classic control environments.

| Transition kernel | Description |
|---|---|
| $\mathsf{P}_{\mathrm{s}}^{s,a}(s_{random_1})$ | The agent moves to a random state $s_{random_1}$ with probability $\alpha r_s$ |
| $\mathsf{P}_{\mathrm{s}}^{s,a}(s_{random_2})$ | The agent moves to a random state $s_{random_1}$ with probability $(1-\alpha)r_s$ |
| $\mathsf{P}_{\mathrm{s}}^{s,a}(s_{other})$ | The agent may move to all other states with probability with probability $\frac{r_s}{|\mathcal{S}|}$. |
| $\mathsf{P}_{\mathrm{t}}^{s,a}(s_{random_1})$ | The agent moves to a random state $s_{random_1}$ with probability $(1-\alpha)r_t$ |
| $\mathsf{P}_{\mathrm{t}}^{s,a}(s_{random_2})$ | The agent moves to a random state $s_{random_1}$ with probability $\alpha r_t$ |
| $\mathsf{P}_{\mathrm{t}}^{s,a}(s_{other})$ | The agent may move to all other states with probability with probability $\frac{r_t}{|\mathcal{S}|}$. |

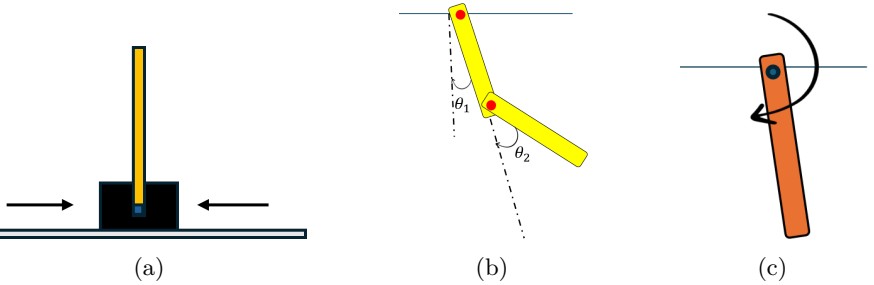

(a)          (b)          (c)

Figure 6: A depiction OpenAI Gym classic control problems: **(a)** Cart Pole. **(b)** Acrobot. **(c)** Pendulum.

**Acrobot environment**: The Acrobot problem, introduced in (Sutton, 1995), features a chain with two links, one fixed and one free end, as depicted in Figure 6b. The objective is to swing the free end of the chain above a certain height, starting from a downward hanging position. The joint between the two links is actuated, allowing for the application of torque to achieve the desired swing. The available actions involve applying a torque to the actuated joint, with options of $\{1, 0, -1\}$ Newton-meter (Nm). Observations in this environment consist of tuples of 6 quantities: $(\cos\theta_1, \sin\theta_1, \cos\theta_2, \sin\theta_2, V_{\theta_1}, V_{\theta_2})$. Here, $\theta_1$ and $\theta_2$ represent the angles of the first link w.r.t. the normal direction and the angle of the second link relative to the first link respectively. The terms $V_{\theta_1}$ and $V_{\theta_2}$ denote the angular velocities of $\theta_1$ $\theta_2$, respectively.

For the observation space, we use a tuple of four quantities $(\cos\theta_1, \cos\theta_2, V_{\theta_1}, V_{\theta_2})$, with a range of $([-1, 1], [-1, 1], [-4\pi, 4\pi], [-9\pi, 9\pi])$, and the number of discrete values $(6, 6, 2, 2)$, respectively. For the reward, we use the quantity $\phi = -\cos(\theta_1) - \cos(\theta_1 + \theta_2)$ to determine different reward levels. Specifically, for $\phi$ falling within predefined ranges ($\phi \geq 1$, $0.5 \leq \phi \leq 1$, $0.25 \leq \phi \leq 0.5$, $0 \leq \phi \leq 0.25$), rewards of 20, 15, 10, 5 are granted, respectively.

**Pendulum environment**: The Pendulum problem is another classic control environment that focuses on the dynamic control of an inverted pendulum. Illustrated in Figure 6c, the system comprises a pendulum affixed at one end to a fixed point. The objective is to apply torque and stabilize the pendulum's center of gravity directly above the pivot, swinging it into an upright position. The action space is continuous, within the range of $[-2, 2]$ Nm. Observations consist of arrays of 3 quantities representing the $x$ and $y$ coordinates of the free end of the pendulum, and the angular velocity, with ranges of $[-1, 1], [-1, 1]$, and $[-8, 8]$, respectively. In our experiment, we discretize the action space into five torque levels $\{-2, -1, 0, 1, 2\}$ Nm. The observation space is segmented into 240 unique states by discretizing the pendulum angle $[-\pi, \pi]$ rad into 12 segments and the angular velocity ranging $[-10, 10]$ rad/s into 20 segments. Rewards are structured to prioritize stabilizing the pendulum near the upright position. Maintaining the pendulum within $[-1, 1]$ of vertical and sustaining a low angular velocity of $[-1, 1]$ earns 100 points. A reward of 50 points is granted for keeping the angle within $[-0.5, 0.5]$, and 10 points are awarded for all other states.

Table 4 provides a description of the transition kernels for both the source and target domains in the classic control problems. For each $(s, a)$ pair, we generate two random states, $s_{random_1}$ and $s_{random_2}$. Taking the source domain as an example, the agent transitions to $s_{random_1}$ and $s_{random_2}$ with probabilities $\mathsf{P}_{\mathrm{s}}^{s,a}(s_{random_1})$ and $\mathsf{P}_{\mathrm{s}}^{s,a}(s_{random_2})$, respectively. For all other states, $s_{other}$, the agent transitions with probability $\mathsf{P}_{\mathrm{s}}^{s,a}(s_{other})$.

We set $r_s = 0.6$, $r_t = 0.7$, and $\alpha = 0.2$ for all environments. To ensure validity, each $\mathsf{P}_s^{s,a}$ and $\mathsf{P}_t^{s,a}$ is normalized to sum to 1.

## E  Additional Experiments

### E.1  Non-robust and robust setting results for the classic control problems.

Figures 7 and 8 show the target domain performance averaged over 20 runs under the non-robust and robust setting, respectively, for the classic control problems.

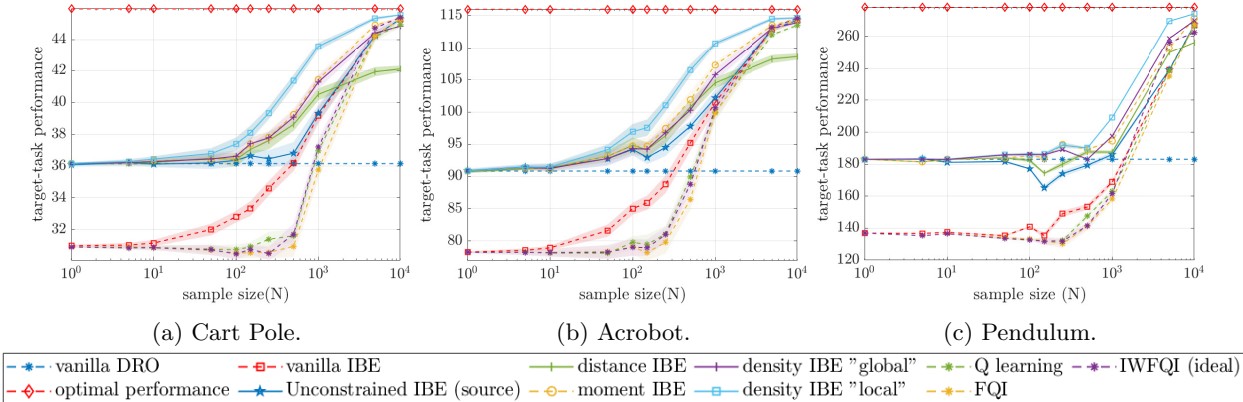

(a) Cart Pole.             (b) Acrobot.             (c) Pendulum.

Figure 7: Target domain performance for the non-robust setting, averaged over 20 runs, as a function of sample size for the classic control problems, and the corresponding 95% confidence interval.

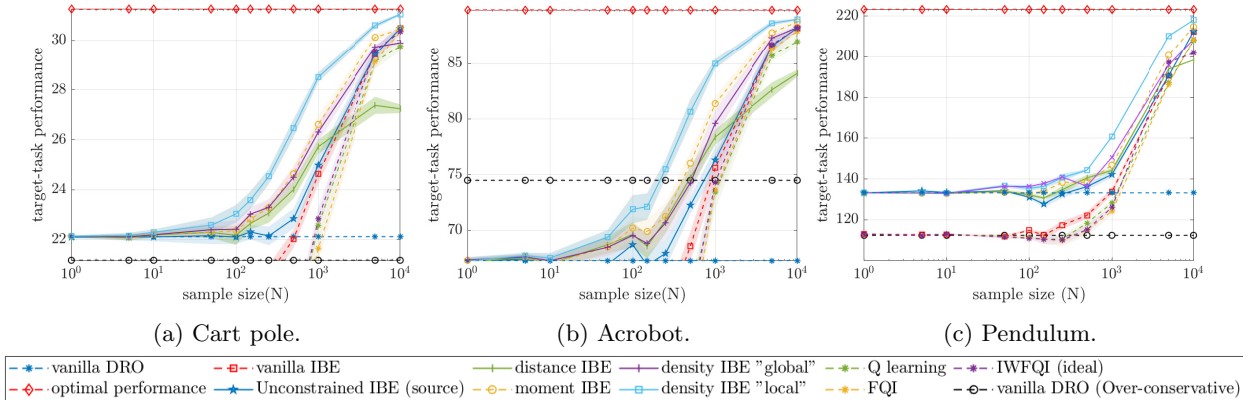

(a) Cart pole.             (b) Acrobot.             (c) Pendulum.

Figure 8: The target domain performance of the robust MDPs averaged over 20 runs as a function of a sample size for the classic control problems, and the corresponding 95% confidence interval.

### E.2  Evaluation error: Experimental verification

In this experiment, we verify the theoretical results of the evaluation error bound presented in Theorem 5 as well as the convergence results of Theorem 6. We consider the frozen lake environment, following the same settings and sampling procedure as in section 5. The policy is estimated by Algorithm 1 using the transition kernels estimated using the approaches outlined in Table 1. Then, each policy is evaluated on the target domain environment. We consider both non-robust and robust scenarios. Figure 9 and 10 show the evaluation (EV) error (RHS of Eq. (11)) and the bound (LHS of Equation 11) as a function of the sample size for each estimation approach for the non-robust and robust settings, respectively. As observed, the evaluation error is

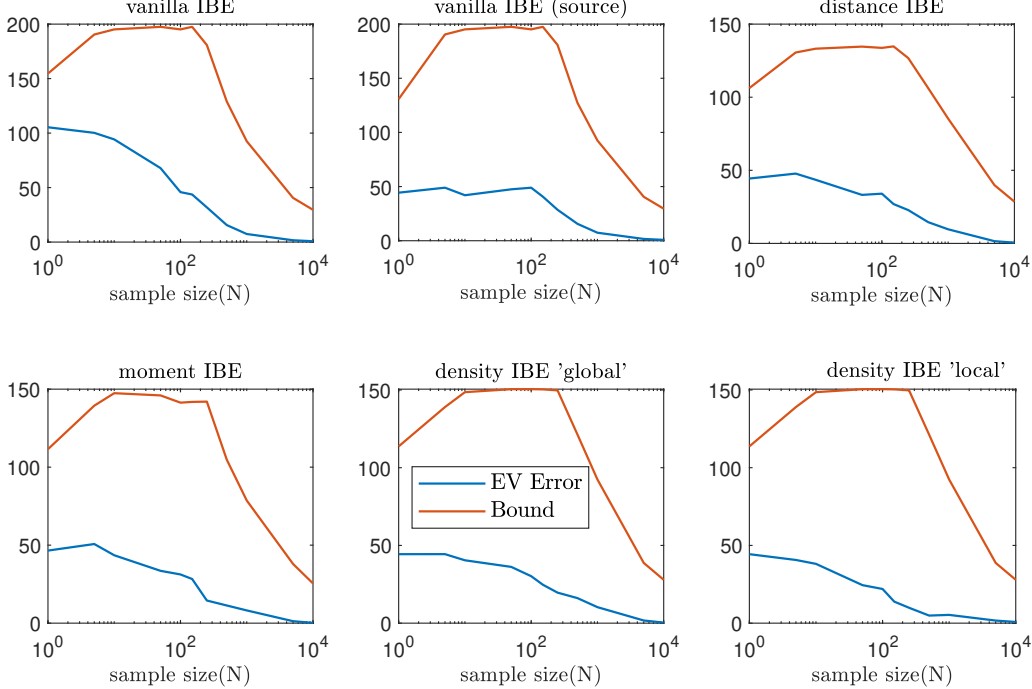

Figure 9: The Evaluation (EV) Error and the bound for each MLE method as a function of the sample size for the non-robust scenario

always upper bounded by the computed bound for all sample sizes. We also observe that the error and the bound decrease as the sample size increases which verifies the convergence results in Theorem 6.

### E.3    The gain of prior information

In this experiment, we investigate the information gain from prior information about the estimated kernels. In particular, given a $k \times k$ FIM $J(\mathsf{P}_\mathsf{t}^{s,a})$, we seek to measure the amount of information that $n$ i.i.d observations provide about the kernel $\mathsf{P}_\mathsf{t}^{s,a}$, given some prior information. Therefore, we consider the following optimization problem:

$$\min_{\mathsf{P}^{s,a} \in \Delta(\mathcal{S})} \operatorname{trace}(J(\mathsf{P}^{s,a})) \quad \text{s.t.} \quad \mathbb{E}^{\mathsf{P}^{s,a}}[x^j] \le c_j, 1 \le j \le M. \tag{43}$$

Here, we assume knowledge of the first $M$ moments. Note that we examine the worst-case scenario for the information the FIM carries by considering the minimum. We compare two scenarios: (i) 'Without Prior', in which we omit the expectation constraints, and (ii) 'With Prior', where the constraints are included.

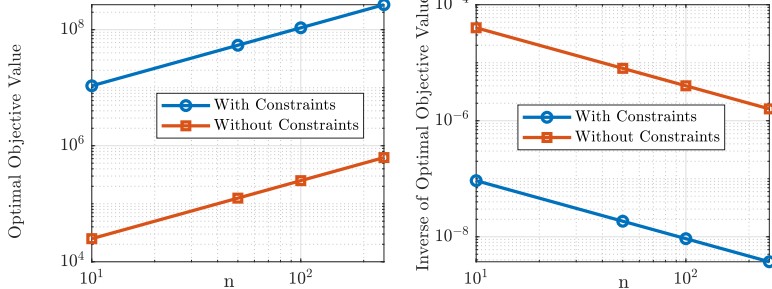

Figure 11: The trace of the FIM as a function of the sample size $n$ (left) and its inverse (right).

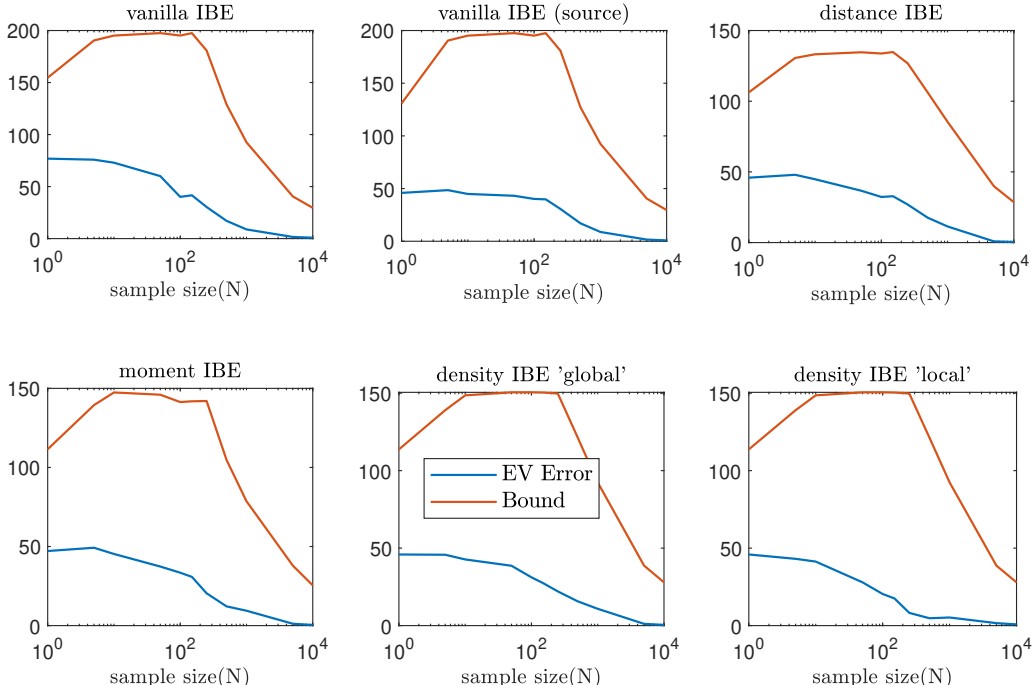

Figure 10: The Evaluation (EV) Error and the bound for each IBE method as a function of the sample size for the robust setting.

Figure 11 shows the optimal value of the trace($J(\mathsf{P}^{s,a})$) (*left*) and its inverse (*right*) on a logarithmic scale. As shown, including the constraints significantly increases the amount of information about the estimator leading to a much better estimate.

