# OpenReview forum: "Robust Policy Adaptation in Markov Decision Processes under Environmental Shift"
_TMLR — Rejected by TMLR_

### Review · Reviewer_fYMD · 2024-11-08

**Summary Of Contributions:**

The authors propose an estimator for transition kernels in Markov decision processes (MDPs). Given a source and a target MDP with different transition kernels, they show how side information about how related the kernels are can be leveraged to get a more accurate estimate of the target kernel from limited samples. The authors provide convergence guarantees for their estimator and, based on the convergence of the estimator, also for the convergence to the optimal policy in the target MDP. Theoretical results are then evaluated in numerical experiments in standard benchmarks and compared to state-of-the-art algorithms.

**Audience:**

Yes

**Claims And Evidence:**

No

**Requested Changes:**

1) Could you explain a bit more intuitively what type of side information you consider? In the introduction and when introducing the concrete objects. What do they say about the environments? How would one obtain this knowledge in practice, or why might it be realistic that it is available?
2) Not necessarily a requested change, but Section 4.1 confused me. I think the motivation was already done in the introduction, I am not sure why it is here needed again.
3) As we have only access to a limited dataset in the target domain, the theoretical results are not that strongly connected to the problem setting. Can you add finite sample guarantees? Or motivate more, why the results are still helpful? It is nice to see that the estimator is consistent, but if it takes forever to converge, it does not help in practice, where I have only a small dataset, that it would converge eventually if I just had more data.
4) Statements like in Theorem 5 tell us that the quality of the estimator is related to the value function. However, as we don't know the true target transition kernel, we also don't know the right-hand side of (10). Can this still help us evaluate the potential quality of our policy?

**Strengths And Weaknesses:**

Strengths
* The estimator, in general, seems reasonable, and the idea of leveraging possible side information makes sense.
* The paper is generally well-written.
* There is sufficient empirical evidence showcasing the benefits of the method.

Weaknesses
* The storyline of the paper is a bit unclear. The main goal is to learn a good policy in a setting where we explicitly have only limited data from the target domain. However, the theoretical results are for the infinite sample limit. How does this relate to the problem setting? This is not discussed
* While it makes sense to consider side information, it is unclear what the considered side information means in practice and how realistic it is that it may be available.
* Apart from convergence, the theoretical results mainly say that more side information can help and that a better estimator implies a better policy. This seems somewhat obvious. But in practice, as we do not know the true kernel, we can also not determine how close we are to the true kernel. So, that result does not really help us evaluate how robust or close to optimal our policy may be.

---

> ### Author Response · Authors · 2024-11-24
> **Response to Reviewer fYMD**
>
> Thank you for your comments and feedback. We have addressed your main concerns in the revised manuscript (section 6: Discussion).   We respond to your comments and questions below.
>
> $\Large{\textbf{Weaknesses}}$:
>
> $ \textbf{The relation between the problem setting and theoretical results} $: Thank you for the question.
>     It is true that only a limited number of samples will typically be available from the target domain. This is precisely why, in our experiments, we have demonstrated that estimating a policy using limited target domain samples, supplemented with side information (i.e., IBE), provides a more accurate estimate of the target distribution compared to vanilla IBE, which relies solely on the available samples. This, in turn, leads to improved policy and value function estimates. Furthermore, we have shown that for varying sample sizes, incorporating side information consistently yields better policies.
>   That being said, studying the asymptotic performance as $n\rightarrow\infty$ remains valuable
>     even with limited samples because it provides theoretical guarantees that our estimates and learned policies are asymptotically consistent, which gives confidence that the method will perform well under ideal conditions. This ensures that even with limited data, the approach is on the right path and will improve as more data becomes available. Also, asymptotic insights often form the foundation for finite-sample analysis, helping to derive practical bounds and understand performance in small-sample regimes. Note that this result supports our main theoretical results on bounding both the training and evaluation errors since convergence of the estimates in total variation implies that these errors approach zero as the sample size increases.
>
> $\textbf{Realism of the side information and its practicability} $: Thank you for the question. It is a valid concern to question the practical realism of the side information we consider. In our work, we use side information in the form of bounds on the TV distance, density ratios, or moment constraints between the source and target transition kernels. These types of information can be realistic in certain practical scenarios. For example, bounds on TV distance can arise naturally in settings with limited domain shifts. For instance, in robotics or control tasks, changes in dynamics such as friction or sensor noise might have quantifiable limits derived from domain knowledge or prior experiments. Density ratio information can be inferred or approximated using a small amount of data from the target domain. This is common in transfer learning applications where target domain data is scarce, but sufficient to estimate relative probabilities of transitions between the domains. Techniques like importance sampling often rely on such ratios. Moment constraints are a common way to encode known properties of the environment. For instance, in physical systems, aggregate or average behaviors are often measurable, even if the full distribution is unknown. Moreover, side information is not limited to the distance, moment, or density function. For example: some $(s,a)$ pairs can be more important than others, in the sense that they capture the similarity between the two environments, in the case of having a different state space. Therefore, the side knowledge carried by these pairs can help in learning an optimal policy on the target domain faster, with a limited number of samples.
>
> The flexibility of our approach allows it to be adapted to a wide range of scenarios where partial knowledge about the relationship between domains is available, which could depend on the application context. By leveraging this information, even when approximate or incomplete, our method achieves significant performance improvements, as shown in our experiments.

---

> ### Author Response · Authors · 2024-11-24
> **Response to Reviewer fYMD**
>
> $\textbf{\Large Weaknesses} $
>
> $\textbf{The absence of target domain transition kernels in practice, and the evaluation of the learned policy}$: The reviewer is correct that, in practice, the true kernel is unknown, and it is challenging to determine how close we are to the true kernel. There are several aspects to consider addressing this concern: (i) Asymptotic Consistency: Our work provides theoretical guarantees on the asymptotic consistency of our IBE for the target transition kernel and the learned policy, as detailed in Theorems 1, 4, and 5. These results underscore that as the sample size increases, our estimates converge to the true kernel, offering reassurance that the performance is improving in the right direction, even if this represents an idealized scenario. (ii) Empirical Evidence: Throughout our experimental analysis, conducted on well-known environments, we demonstrate that our approach outperforms other baseline methods with finite sample sizes. This highlights its practical utility and effectiveness even when asymptotic conditions are not met. (iii) Robustness to Perturbations: In the robust setting, we evaluate performance with respect to uncertainty sets centered around the true kernel. This approach accounts for potential perturbations in the true kernel itself and evaluates the worst-case performance of our method compared to others under the same uncertainty set. This addresses uncertainty in the target domain kernel. (iv) Challenge of Unknown Optimal Kernel: The issue of not knowing the true optimal policy or kernel is inherent to most RL problems. To address this, we test our approach on well-known environments where the optimal policy is known or can be approximated. This allows us to evaluate how well our approach performs relative to the optimal policy, in addition to providing theoretical guarantees. In summary, while the true kernel is unknown in practice, our theoretical results, empirical evidence, and robust evaluation provide strong support for the effectiveness and reliability of our method.
>
> $\textbf{\Large Requested Changes} $
>
> 1. $\textbf{Insights on the side information and its realism in practice} $:In our work, we use side information in the form of bounds on the TV distance, density ratios, or moment constraints between the source and target transition kernels. These types of information can be realistic in certain practical scenarios. For example, bounds on TV distance can arise naturally in settings with limited domain shifts. For instance, in robotics or control tasks, changes in dynamics such as friction or sensor noise might have quantifiable limits derived from domain knowledge or prior experiments. Density ratio information can be inferred or approximated using a small amount of data from the target domain. This is common in transfer learning applications where target domain data is scarce, but sufficient to estimate relative probabilities of transitions between the domains. Techniques like importance sampling often rely on such ratios. Moment constraints are a common way to encode known properties of the environment. For instance, in physical systems, aggregate or average behaviors are often measurable, even if the full distribution is unknown.  The flexibility of our approach allows it to be adapted to a wide range of scenarios where partial knowledge about the relationship between domains is available, which could depend on the application context. By leveraging this information, even when approximate or incomplete, our method achieves significant performance improvements, as shown in our experiments.
>
> 2. $\textbf{Why we added a motivation section, section 4.1}$:
> The reason we added Section 4.1 is to clearly articulate the motivation behind the proposed approach after defining the exact problem setup at the end of Section 3. Specifically, we argue that an offline RL approach is feasible but would require a large sample complexity, which is later evidenced in our experiments when we compare to vanilla IBE. Using the source MDP directly, however, would lead to degraded performance if there is a significant discrepancy between the transition kernels of the two MDPs. Also, applying standard robust RL would result in an over-conservative policy that relies solely on source information. Our proposed method addresses this issue by remedying the over-conservative solution, where the source transition kernels serve as the center of the uncertainty set, by constructing a smaller uncertainty set centered around the estimated kernels. These estimated kernels are closer to the target dynamics than the source kernels, leading to improved performance.
>     We consider this motivation essential to include immediately after the problem formulation in Section 3 to ensure a logical progression and a clear rationale for the proposed method. With the aid of Figure 1, we believe the underlying idea of our approach is conveyed to the reader more smoothly and clearly.

---

> ### Author Response · Authors · 2024-11-24
> **Response to Reviewer fYMD**
>
> $\textbf{\Large Requested Changes} $
>
> 3. $\textbf{ The connection between the theoretical results and the problem setting:} $
> The reviewer is correct that deriving finite sample guarantees or characterizing exact rates of convergence would be highly valuable, but this is a challenging problem and is beyond the scope of the current study. Addressing this would require substantially more analytical machinery and theoretical development, extending beyond the already under-studied topic of finite sample complexity in offline robust RL.
>     However, in our experiments, we have demonstrated that estimating a policy using limited target domain samples, supplemented with side information (i.e., IBE), yields a more accurate estimate of the target distribution compared to vanilla IBE, which relies solely on the available samples. This, in turn, leads to improved policy and value function estimates. Moreover, for varying sample sizes, we have shown that incorporating side information consistently results in better policies.
>     That said, studying asymptotic performance remains valuable, even with limited samples. In particular, it provides theoretical guarantees that our estimates and learned policies are asymptotically consistent, offering confidence that the method is progressing in the right direction and will improve with more data. Furthermore, asymptotic results often form the foundation for finite-sample analysis, helping to derive practical bounds and understand behavior in small-sample regimes. The asymptotic results support our main theoretical contributions, which bound both training and evaluation errors, since convergence of the estimates in total variation ensures these errors approach zero as the sample size increases.
> We emphasize that investigating finite sample guarantees is of significant interest to us, as the reviewer pointed out. However, this endeavor would require a much deeper analysis that is currently beyond the scope of this study. A potential approach would be to use some concentration inequalities to develop finite sample analysis for our approach. We plan to explore this direction in future work.
>
> 4. $\textbf{ Evaluating the RHS of (10), and the absence of the target domain transition kernels} $:
>  It is true that we do not know the true target transition kernel, and therefore we cannot directly compute the RHS of (10). However, the relationship established in Theorem 5 remains valuable for evaluating the potential quality of our policy as it provides a theoretical connection between the total variation distance of the estimated and true target kernels and the suboptimality gap of the value function. This guarantees that as the kernel estimates improve (e.g., with more samples and/or better side information), the learned policy approaches the optimal policy. While we cannot observe the true kernel, this gives confidence that our method is on the right track. Also, although the exact RHS of (10) is unknown, our experiments demonstrate that the policies learned through our approach consistently outperform those learned through baseline methods such as vanilla IBE. This indicates that incorporating side information reduces the total variation distance and improves the policy quality, even without direct access to the true kernel.
>     Therefore, although the true target kernel and the exact RHS of (10) are unknown, the insights from Theorem 5, combined with empirical evidence, provide a strong basis for understanding and evaluating the potential quality of our policy.

---

> > ### Author Response · Authors · 2024-11-26
> > **Response to Reviewer fYMD**
> >
> > We sincerely appreciate the reviewer’s time and effort in evaluating our manuscript. If there are any further concerns, please do not hesitate to share them as the discussion period approaches its conclusion.

---

> > > ### Comment · Reviewer_fYMD · 2024-11-27
> > >
> > > I thank the authors for their explanations and changes in the paper. The discussion section does help in putting things into perspective. One could consider moving the future work discussion into the conclusions, but that is obviously nothing critical.
> > >
> > > The extended discussion on the side information is helpful. In the experiments, it is still not entirely clear to me how the side information was obtained. Is this based on the true transition kernel? Since this is assumed unknown, I think it would be a good possibility to discuss in practice how one would obtain the side information from an unknown environment by, e.g., estimating it. Or discussing for one of the examples, what concrete knowledge is required to obtain it.

---

> ### Author Response · Authors · 2024-11-28
> **Response to Reviewer fYMD**
>
> Thank you for your time and effort in reviewing our work.
>
> $\textbf{Side information in our experiment and in practice.}$
> In our work, we assumed that the side information is provided (i.e., known). In practical scenarios, such information can often be inferred or measured. For example, bounds on total variation (TV) distance may naturally emerge in settings with limited domain shifts. In robotics or control tasks, for instance, changes in dynamics—such as variations in friction or sensor noise—can often be constrained using domain knowledge or prior experimental data. Similarly, density ratio information can be inferred or approximated using a small sample of data from the target domain. This is particularly common in transfer learning applications, where target domain data is limited but sufficient to estimate relative probabilities of transitions across domains. Techniques like importance sampling frequently leverage such ratios. Additionally, moment constraints are a widely used method for encoding known properties of the environment. For example, in physical systems, aggregate or average behaviors are often measurable, even when the full underlying distribution is unknown. We have included these insights in the newly added discussion section preceding the paper’s conclusions.

---

### Review · Reviewer_6tSu · 2024-11-13

**Summary Of Contributions:**

The paper proposed a policy adaptation method based on the information based estimation for domain adaptation in RL. The problem setting took the idea of uncertainty set in robust RL and showed that the proposed method can converge asymptotically if the environmental shift can be quantified by the uncertainty set.

**Audience:**

Yes

**Broader Impact Concerns:**

No concerns about the broader impact. The work concerns about domain adaptation in RL, which is potentially beneficial to the society in various applications.

**Claims And Evidence:**

Yes

**Requested Changes:**

I would suggest adding at least an example (maybe with a different uncertainty set) to illustrate that robust RL method is over conservative but the proposed method is not.

**Strengths And Weaknesses:**

The formulation is interesting and reflects a lot of cases in real life application. This formulation is also important as the existing robust RL method provides an over-conservative solution. While I believe the proposed method can potentially be really useful in the field, it is unclear to me how it should be compared to a robust RL method (from a theoretical standpoint).

Intuitively, as the robust RL method are over-conservative, its performance should degrade significantly with $R$ while the proposed methods is less affected. However, with the current choice of uncertainty set (which is described by TV distance), and the asymptotic bound, this is not reflected. Part of this could be due to the fact that robust RL method is not really affected by the uncertainty set radius when it is given by TV distance. So I am not sure if TV distance would be a good choice of evaluating the proposed solution.

---

> ### Author Response · Authors · 2024-11-24
> **Response to Reviewer 6tSu.**
>
> We would like to thank the reviewer for the comments and feedback.
>
> $\Large{\textbf{Weaknesses}}$:
>
> $\textbf{Comparison to a robust RL method (from a theoretical standpoint)}$: In our work, we primarily focus on two objectives. First, we provide theoretical guarantees on the asymptotic convergence of our estimates (IBE), leveraging side information about the transition kernels to demonstrate the consistency of our estimator. Second, we establish asymptotic guarantees and derive an upper bound on the evaluation and training errors, which depend on the TV distance between the IBE and the true target transition kernel. We believe these contributions are significant for understanding the asymptotic behavior of the learned policy given our IBE.
> Existing work in robust RL does not incorporate side information or consider transfer between two different MDPs. In this context, a straightforward robust RL approach corresponds to the over-conservative case we compare against, where the uncertainty set is constructed around the nominal kernel—in this case, the source. In contrast, we leverage side information to build an alternative uncertainty set around an estimated kernel that integrates information from both the source and the target domains. To the best of our knowledge, this approach is introduced here for the first time.
>
> $\textbf{Over-conservative case and the uncertainty set's radius}$: Whether robust RL performance degrade significantly with $R$ depends on the environments. For instance, it is possible that with different radius $R$, the worst-case remains similar, and hence perform similarly.
>  In our robust setting, the over-conservative policy is learned using robust value iteration with an uncertainty set centered at the source domain transition kernel and a radius of
>  $R$ +$ TV(\mathsf{P}^{s,a}_s, \mathsf{P}^{s,a}_t )$, where $TV$: is the total variation distance
> and R = 0.1, as discussed in Section 5.3. This policy is then tested on the target domain using Algorithm 2, which uses robust value function iteration to evaluate the learned policy on the target domain uncertainty set with a radius of $R = 0.1$ centered at the target transition kernel. The radius of the uncertainty set for the over-conservative case is chosen to ensure it includes the target domain test set and is not a function of the sample size. Consequently, the performance of the over-conservative policy should remain unaffected by the sample size.
> The primary goal of this experiment is to demonstrate that our policy outperforms the over-conservative policy when tested on the target domain. Additionally, while the performance of the learned policy for different radii is not the central focus of our theoretical results on error bounds, the experiment sheds light on the relationship between the value function error and the total variation distance between the estimated and true target kernels. As established in Theorems 4 and 5, the value function error is proven to be upper bounded by this distance. Therefore, as the transition kernels converge in total variation, we can drive both evaluation and training errors to zero.
>
> $\Large{\textbf{Requested changes}}$:
>
> $\textbf{ An experiment for the over-conservative case:}$
> The over-conservatism lies in expanding the radius of the uncertainty set to include the target domain test set. We have considered this comparison in our numerical experiments in Section 5.3. Specifically, since we have a bound on the distance between the kernels, we used an uncertainty set centered at the source kernel, with the radius enlarged to include the target domain uncertainty set (the test set), i.e., $\text{Radius} = R + TV(\mathsf{P}^{s,a}_s,\mathsf{P}^{s,a}_t)$.
>
> We needed to enlarge the uncertainty set in this way to have guarantees on the value function of the optimal policy. In particular, the performance on the target domain is better than the worst-case performance. However, this approach defends against a broad range of distributions, many of which are not relevant, thereby degrading performance on the target domain distribution that we care about.
>
> As observed in our experiments, this over-conservative case performed poorly in the target domain environment. In contrast, our IBE approach, which leverages limited data from the target domain along with side information, outperformed the over-conservative approach. Furthermore, as more samples are available from the target domain, the policy learned through our method exhibits increasingly better performance in the target domain.

---

> > ### Author Response · Authors · 2024-11-26
> > **Response to Reviewer 6tSu**
> >
> > We sincerely appreciate the reviewer’s time and effort in evaluating our manuscript. If there are any remaining concerns, please feel free to let us know as the discussion period concludes.

---

> > > ### Comment · Reviewer_6tSu · 2024-11-28
> > >
> > > Thank you for the response.
> > >
> > > I think the added experiment helps a lot in understanding the effect of the IBE-type methods. However, I still want to ask whether we can see a difference between the theoretical performance of robust RL and the IBE methods when the uncertainty radius is not measured by TV. I don't think either method's performance is affected by TV, and that may just be because of the nature of TV. Would other distance matrics work the same way? I would not expect rigorous proof on this, but a high-level explanation would really help.

---

> > > > ### Author Response · Authors · 2024-11-28
> > > > **Response to Reviewer 6tSu**
> > > >
> > > > Thank you for reviewing our work and providing additional feedback.
> > > >
> > > > We appreciate the insightful observation, as your intuition aligns closely with ours. In our study, we focused exclusively on the TV distance as the metric for defining uncertainty sets and developed theoretical results regarding the convergence of IBE with various types of side information, along with error bounds that primarily depend on the TV distance. However, as you correctly point out, we believe that similar insights could extend to other metrics for defining uncertainty sets, such as Chi-square divergence, Wasserstein distance, and others.
> > > >
> > > > That said, for uncertainty sets defined using metrics other than the TV distance, a more detailed and tailored analysis would be required. This would include: (i) analyzing the convergence of the estimates with respect to the chosen metric, (ii) deriving error bounds and their dependence on that metric, and (iii) developing corresponding policy estimation and evaluation algorithms.
> > > >
> > > > Error bounds could be more challenging for some uncertainty sets due to the more intricate dependence of the support function on the kernel. For example, for a Chi-square divergence uncertainty set, the support function is given by: $\sigma_\mathcal{P}(V)=\max_\mu \{\mathsf P^{s,a} (V-\mu) - \sqrt{R\operatorname{Var}_{\mathsf P^{s,a}}(V-\mu) }\}$, where the variance term introduces additional non-linear dependencies on the nominal kernel. This complexity makes the analysis more involved compared to TV distance.
> > > >
> > > > Nonetheless, we agree in principle with the idea that as the estimated transition kernel approaches the true target kernel, the estimated policy performance should improve, irrespective of the specific metric used to define the uncertainty set. However, rigorous theoretical validation would be necessary to establish the applicability and behavior of IBE methods under alternative metrics.

---

### Review · Reviewer_Bjbb · 2024-11-14

**Summary Of Contributions:**

This paper tries to tackle the challenge of environmental shift between a “source” and a “target environment” due to differences in the transition kernels, by developing a method based on info-based, avoiding over-conservative policy. They assume the following:

1. Access to the source transition kernel.
2.  Access to a few samples from the target environment (s, a, s’).
3.  Prior knowledge about the relationship between source and target transition kernel.

**Paper contributions**:

1. Introduce an info-based approach to estimate the target environment transition kernels using side information from the source environment, and prior knowledge between the target and source environments.
2. Proof that the estimate of the target kernel transition asymptotic converges to the true transition w.r.t total variation distance, under some assumptions.
3. Proof error bounds for value function in robust and non-robust settings.
4. Empirical experiments show that the info-based method, with prior info, outperforms the state-of-the-art baselines in toy and control environments.

**Audience:**

Yes

**Claims And Evidence:**

No

**Requested Changes:**

1. Describe how transition kernels are ”related” but “differen”t between the source and target environments, especially in the experiments.
2. Maximum likelihood estimator (MLE) may fail to exist, how does the proposed approach deal with this? Also, MLE may not be unique, and multiple estimators may exist, how does the proposed approach handle that? How to estimate the uncertainty for MLE?
3. How non-robust environments, R=0.1 is it standard deviation noise added to the next state probability?
4. What is the y-axis in Figure 2? Is it the greedy policy performance after training (evaluate)? Or value function of the initial state $V(s_{0})$? Or something else?
5. Is it fair to compare Q-learning, which does not have access to the prior knowledge about transition kernels, to other methods that have access to this prior? Is it fair to compare these baselines with differences in the computation resources?
6. As the number of samples from the target environment increases, the performance of all baselines increases, are all samples equal? Maybe some (s, a, s`) pairs are more important than others, perhaps explaining the difference between all the source and target transition kernels.
7. Is the “average total variation distance” a good measure of how close the estimate and true target transition kernels are?
8. What will happened if the prior info $\phi(P, P_{s})$ is not accurate, e.g., distance IBE if d=1 → $|| P_{t}^{s, a} - P_{s}^{s, a} ||_{1} = 1$? Is it a useful info?
9. In density IBE: is it realistic to assume that “the target distribution is absolutely continuous w.r.t. The source distribution”?
10. Notation table to avoid confusion, and use unique notation, e.g. in Section 4.3 $V^{\pi_{n}}{P_{n}}$ is confusing.

11. Figures:
Figure 1: Explain the y-axis, which represents the average distance, is it the average probability over all state-action-next state pairs? $\sum_{\hat{s} \in S} Pr(\hat{s} | s, a)$ or max distance for all state-action-next state pairs?
Figure 1: Add a confidant interval over 20 runs.
Figure 1: clip the y-axis to start at zero.
Figure 1: Add sub-captions with environment names.
Figure 2: separate legends into a new sub-figure.
Figure 2: Clip the y-axis (max) with the optimal policy reward.
Figure 2: Add sub-captions with environment names.
Figure 2: increase the figure size or reduce the number of graphs, having 11 curves in a small figure makes it hard to highlight findings.
Figure 2: Add sub-captions with environment names.
Figure 3: separate legends into a new sub-figure.
Figure 3: Add sub-captions with environment names.
Figure 3: Add a confidant interval over 20 runs.
Figure 3: increase the figure size or reduce the number of graphs, having 12 curves in a small figure makes it hard to highlight findings.

12. Typos:
     1. Equation 3: $\eta = (p_{0}, p_{1}, …)$ not   $\eta = (p_{0}, \hat{p}_{1}, …)$.
     2. Figure 1 caption: c) not c):
     3. Subsection 5.3: Figure 4 present**s** not Figure 4 present

**Strengths And Weaknesses:**

**Strengths**:
1. Environmental shift is an important and interesting problem to study.
2. Related works cover multiple areas that look into similar problems, from multitask RL, and domain adaptation, to robust RL and domain randomization.

**Weaknesses:**
- Evidence supporting the claim is unclear.
    1. Contribution 1: clear contribution.
    2. Contribution 2: they assume $n \rightarrow \infty $, however, this is an unrealistic assumption since there are a few samples from the target environment. So there is a big gap between the claim and the evidence presented.
    3. Contribution 3: **I do not have a strong background on driving error bounds**, but is built upon contribution 2, which makes it harder to validate.
    4. Contribution 4: empirical results hint that the IBE could converge, however, in all environments the every distance to the true target is not zero in Figure 2, even with $10^4$ samples for environment where number of state-action pair are 64, 152, and 648 respectively. But the policy performance is very close to optimal in non-robust (Figure 3) and robust (Figure 4).
2. There is no description of how the source and target environment are different in their transition kernels, even after checking the appendix.
3. Lake of a description of how the prior knowledge implies inside the algorithm.
Missing the definition of “environmental shift”? How does “mismatch transition” differ from sharing a common transition between source and target environments?
4. Theorem 1 is “stating the obvious”, as the number of samples from the target environment → inf, the distance between the estimator and true distribution = 0, so it should be a lemma.
One of the submission goals is to learn the optimal policy for the target MDP with and without uncertainty, however, the performance of the proposed approach does not reach the optimal performance (it is close) in Figures 3 and 4.
5. Results representation could be misleading:
    1. Although there are 20 runs for all experiments, all graphs report the average only, which is insufficient, unless the average is combined with a confidence interval, standard deviation, or other statistical measure for the variation across runs.
    2. crowded graphs, in a small sub-figure, shows more than 12 curves with a legend, and small
    3. The y-axis in Figure 2, which shows the average distance, should be clipped to 0 for all environments.
6.  Some equations are not clear, for example, equation 4: $n_{j}$

---

> ### Author Response · Authors · 2024-11-23
> **Response to Reviewer Bjbb :**
>
> Thank you for your comments and feedback, we have addressed your comments in the revised manuscript. Below, we give point by point response to your comments and questions.
>
> $\Large{\textbf{Weaknesses ("Evidence supporting the claim is unclear")}: }$
>
> 2. $\textbf{Contribution 2}$ : The reviewer is correct that in practice, only a limited number of samples will typically be available from the target domain. This is precisely why, in our experiments, we have demonstrated that estimating a policy using limited target domain samples, supplemented with side information (i.e., IBE), provides a more accurate estimate of the target distribution compared to vanilla IBE, which relies solely on the available samples. This, in turn, leads to improved policy and value function estimates. Furthermore, we have shown that for varying sample sizes, incorporating side information consistently yields better policies.
>      That being said, studying the asymptotic performance as $n\rightarrow\infty$ remains valuable
> even with limited samples because it provides theoretical guarantees that our estimates and learned policies are asymptotically consistent, which gives confidence that the method will perform well under ideal conditions. This ensures that even with limited data, the approach is on the right path and will improve as more data becomes available. Also, asymptotic insights often form the foundation for finite-sample analysis, helping to derive practical bounds and understand performance in small-sample regimes. Note that this result supports our main theoretical results on bounding both the training and evaluation errors since convergence of the estimates in total variation implies that these errors approach zero as the sample size increases.
>
> 3. $\textbf{Contribution 3}$: We have included such validation of error bounds in Appendix E.2 of the revised manuscript using toy-text environments. To clarify, in Contribution 3, we establish asymptotic performance guarantees for the policies learned under our models, demonstrating that our methods yield a near-optimal policy in the target domain. Specifically, we prove that the sub-optimality gap between our policies and the optimal policy in the target domain depends on the total variation distance between the estimated and true target kernels. Building on our earlier result on the asymptotic consistency of the estimates, this demonstrates the asymptotic convergence of our policies to the optimal policy, which is a main theoretical contribution of this work. Notably, this result holds for both the robust and non-robust settings.
>
> 4. $\textbf{Contribution 4}$: Thank you for this question. First, in Figure 2, we report the average total variation (TV) distance across all state-action pairs, averaged over 20 runs. Second, note that $N$ represents the total number of samples $(s,a,s')$, so coverage across all state-action pairs is not uniform—state-action pairs with low probability are less well-sampled. Reducing the transition error for each state-action pair to zero would require a large sample size for each pair. Third, and importantly, Theorem 5 provides an upper bound on the error in terms of this distance, meaning it is only an upper bound. Even if there is a gap in the distance, the suboptimality gap can still be small. Indeed, recent work on the sample complexity of model-based offline RL [1,2] (in which we learn an optimal policy from a dataset pre-collected using a behavior policy) demonstrates that full coverage of all state-action pairs is not necessary; we only need coverage for state-action pairs visited by the optimal policy. Consequently, even if estimation is poor for less critical state-action pairs, it does not hinder the identification of the optimal policy. Optimal policies can still be learned effectively, even under partial coverage with less explored datasets for which the total TV distance of the estimated kernel to the true target could be relatively large.\\
>
> [1] Paria Rashidinejad, Banghua Zhu, Cong Ma, Jiantao Jiao, and Stuart Russell. Bridging offline reinforcement learning and imitation learning: A tale of pessimism. Advances in Neural Information
> Processing Systems, 34:11702–11716, 2021.\\
>
> [2] Laixi Shi and Yuejie Chi. Distributionally robust model-based offline reinforcement learning with
> near-optimal sample complexity. arXiv preprint arXiv:2208.05767, 2022.

---

> ### Author Response · Authors · 2024-11-23
> **Response to Reviewer Bjbb**
>
> $\Large{\textbf{Weaknesses}}$:
>
> 2. $\textbf{Description of the source and target transition kernels}$: We have added a description in the Appendix of the revised version (See Table 3 and
> Table 4 and the added texts). We generate the transition kernels for each environment such that
> they are relatively distant (e.g., TV distance between 0.4 and 0.7).
>
> 3. $\textbf{Prior information in the algorithm, and the environment shift}$: The prior information is used in the first step of our algorithm to obtain the estimate
> (IBE). This information is provided as distance, density ratio, or moment constraints that relate
> the source and target kernels. We defined the environment shift in Section 3. We assume the two
> environments share the same state, action spaces and reward but differ in their transition dynamics.
> The prior information, expressed through these constraints, captures the relationship between the
> differing dynamics of the two domains.
>
> 4. $\textbf{The distance convergence and the optimal performance in Figures 3 and 4 }$:
> The convergence result states the convergence with prior information in the form of
> constraints on distances, density ratios, or moments, which is not a direct result.
> A major goal of our experiments is to demonstrate that incorporating side information about the
> relationship between domains significantly outperforms vanilla IBE, which estimates the transition
> kernel using offline data alone, without utilizing such information. In many cases, our approach
> successfully reaches the optimal policy of the target transition kernel. However, achieving this in
> some environments may require a larger number of samples. Performance also depends on the quality
> and type of side information provided. For instance, density IBE with "local" side information
> demonstrates the fastest convergence to the optimal policy based on the true target domain kernels.
>
> 5. $\textbf{ Presentation of the results} $: Following your suggestion:(ii) we have included the 95% confidence interval to the graphs.
> (ii) We have increased the size of the figures and separated the legend for better readability.
> (iii) The y-axis has been clipped to $0$ in Figure 2.
>
> 6. $\textbf{Equation 4 and $n_{j}$}$:
> Equation (4) states that the optimal policy is the one that maximizes the worst case value function, where the worst-case is over the uncertainty set of distributions, which is standard in the robust MDP literature.
>
> Consider the state space $\mathcal{S} = \{1, 2, \ldots, K\}$. Let $n$ be the total number of samples drawn from the $(s,a)$ target domain transition kernel $\mathsf{P}^{s,a}_t $.
>
> For each state $j \in \mathcal{S}$, $n_j$ represents the number of samples among the $n$ total samples such that the next state $s' = j$.  Formally, $ n_j$ = $\sum_{i=1} ^n \mathbb{I} [{s'} _i = j] $,
>
> where $\mathbb{I}[\cdot]$ is the indicator function. The total samples satisfy the constraint $\sum_{j=1}^K n_j = n$. This partitioning of $n$ into $n_j$ values represents the empirical distribution of next states under the target domain transition kernel $\mathsf{P}^{s,a}_t$. We have clarified this in the revised manuscript.
>
> $\Large{\textbf{Requested changes:}}$:
>
> 1. $\textbf{ "related" but "different" source and target environments} $:  We assume the two environments share the same state and action spaces but differ in their transition dynamics. The prior information, expressed through these constraints, captures the relationship between the differing dynamics of the two domains.
>
> 2. $\textbf{ Existence and Uniqueness of the MLE} $: The MLE may fail to exist in certain cases, such as when the likelihood function is unbounded or ill-behaved due to discontinuities. However, this is not an issue in the settings we consider. Specifically, the likelihood function, given by, $\max_{\mathsf P^{s,a} \in \Delta(\mathcal{S})} \sum_{j=1}^{|\mathcal{S}|} n_j \log \mathsf P^{s,a}_{j}$, is a well-behaved and proper function. Moreover, our IBE is guaranteed to be unique because the objective (the likelihood function) is strictly concave, and the constraints under consideration are convex. For further details, please refer to the proof of Theorem 1.
>
> 3. $\textbf{ $R$ in the non-robust setting}$: In the non-robust case, the radius $R=0$, recalling that $R$ is the radius of the uncertainty set. In this case, we use value function iteration to estimate the policy and evaluate it on the target domain with $R=0$.

---

> ### Author Response · Authors · 2024-11-24
> **Response to Reviewer Bjbb**
>
> $\Large{\textbf{Requested changes}}$:
>
> 4. $\textbf{ y-axis in Figure 2}$:In Figure 2, the y-axis represents the average Total Variation (TV) distance between the estimated and target domain transition kernels averaged over all $(s,a)$ pairs. We reported the results, averaged over $20$ runs. If you are referring to Figure 3 or Figure 4, yes, the y-axis represents the evaluated performance of the learned policy after training. More specifically,  we use Algorithm 2 (See Appendix C) to evaluate our learned policy for the non-robust and robust scenarios, which correspond to Figure 3 and Figure 4, respectively. We computed the value function $V^{\pi}(s), s\in \mathcal{S}$ ,where $\pi$ is the learned policy, and reported the average over all states, i.e. $\frac{1}{|\mathcal{S}|}\sum_{s\in \mathcal{S}} V^{\pi}(s)$, on the y-axis. The results are averaged over $20$ runs.
>
> 5. $\textbf{Comparing to Q-learning and other baseline methods} $: Our primary goal in considering Q-learning and other offline methods, such as FQI and IWFQI, which rely solely on target domain samples, is to highlight the advantage of leveraging side information and, consequently, the power of transfer from a related domain under domain shift. This is evidenced by the superior performance of the policies estimated using our approach in the target domain. While these baseline methods vary in their computational requirements—some, like IWFQI, being specifically designed to address the environment shift problem—our approach consistently outperforms them.
>
> 6. $\textbf{Number of target domain samples, and the performance for different sample sizes} $: Yes, for each sample size $N$, we use the same samples across all baseline methods. As previously explained, $N$ represents the number of tuples $(s, a, s')$ sampled from the target domain transition kernel, but the coverage of state-action pairs is not uniform. In our approach, the $(s, a)$ pairs that are covered are those for which the transition kernels will be estimated, while the transition kernels for the remaining $(s, a)$ pairs are set to the corresponding values from the source transition kernel, as detailed in Section 5.
>
>     As more samples become available for a specific $(s, a)$ pair, the estimates of the transition kernel improve, as established in Theorem 1. As mentioned in our response to your earlier question, certain state-action pairs might indeed be more critical, particularly if they are visited by the optimal policy. However, since the optimal policy is unknown, the sampling process cannot prioritize these pairs. Instead, the samples are typically collected using a uniform or another behavior policy, as is standard in offline RL.
>
> 7. $\textbf{Using the average total variation distance as an evaluation metric} $: Since we use TV distance to define the uncertainty sets, we also use TV as a measure of how close the transition kernels are for compatibility. Our objective, as illustrated in Section 4.1, is to obtain a better estimate of the target domain transition kernels and then estimate a policy that performs well in the target domain under uncertainty. The uncertainty set centered around the estimated kernels requires a smaller radius to include $\mathsf{P}_{\text{t}}$ compared to the over-conservative case where the uncertainty set is centered around the source distribution. This uncertainty set, defined with respect to TV distance, is then used to obtain the policy.
>
> For example, consider an uncertainty set $\mathcal{P}$ defined with respect to the TV distance with radius $R$ and centered at the estimated kernel $\hat{\mathsf{P}}$. Let $\pi^*$ denote the estimated policy such that $TV(\mathsf{P}_t, \hat{\mathsf{P}}) \leq R$.
>
> Then, it is guaranteed that the value function satisfies:
>
> $V^{\pi^*}_{\mathsf{P}_t}(s) \geq$
>
> $V^{\pi^*}_{\mathcal{P}}(s), \quad \forall s \in \mathcal{S}$,
>
> which provides a lower bound on the value function evaluated at the true $\mathsf{P}_{\text{t}}$. Thus, using the TV metric offers insight into the quality of the policy.
>
> Additionally, since there are $|\mathcal{S}| \times |\mathcal{A}|$ transition kernels, averaging the TV distances over these transitions is a reasonable metric. We have also included $95$ percent confidence intervals in the figures, as requested.

---

> > ### Author Response · Authors · 2024-11-24
> > **Response to Reviewer Bjbb**
> >
> > $\Large{\textbf{Requested changes}}$:
> >
> > 8. $\textbf{What if the prior information is inaccurate}$: In the case you are mentioning, the side information would not be useful. Naturally,  if the quality of the side information is low, the prior information may provide little to no benefit. In our work, we explored various types of side information, including density, moment, and distance constraints. For our method to succeed, the side information must be sufficiently informative to reveal meaningful relationships between the two environments.
> >
> >     As the quality of the side information improves, it leads to better policy estimation. This is evident in our experimental results, where we observe that density ratio information significantly aids in learning a good policy for the target domain, even with a relatively small number of samples.
> >
> > 9. $\textbf{The realism of the absolute continuity assumption in density IBE}$: The absolute continuity assumption, while idealized, is a common and practical modeling assumption in transfer learning and domain adaptation contexts. In many real-world scenarios, the source and target domains often share significant overlap in their dynamics, particularly if they represent related environments. This overlap makes the assumption reasonable in practice.
> >
> >     However, we acknowledge that there can be cases where the assumption is violated for certain states or transitions. In such situations, techniques like assigning a small value $\epsilon$ to ensure compatibility provide a practical workaround. These adjustments allow the model to remain applicable while acknowledging the potential sparsity or divergence in certain parts of the state-action space. Overall, while not universally valid, the assumption is sufficiently realistic for a wide range of applications where domain relationships exist.
> >
> > 10.  $\textbf{Notation table, and}$ $V^{\pi_{n}} \mathsf{P}_{n}$  $\textbf{in Section 4.3}$: Per your suggestion, we have included a notation table in the appendix of the revised manuscript.
> >
> > $V^{\pi_n}_{\mathcal{P}_n}$
> >
> > refers to the worst-case value function under the policy $\pi_{n}$, evaluated over the uncertainty set $\mathcal{P}_n$. This uncertainty set, estimated for a sample size $n$, consists of all distributions that are within a specified distance from the estimated kernel, with a cross-product structure over the kernels for all $(s, a)$ pairs.
> >
> > 11. $\textbf{Figures}$:
> >
> >   We believe that you are referring to Figure 2. The y-axis of Figure 2 represents the average TV distance between the estimated and target domain transition kernels, $\hat{\mathsf P}^{s,a}$ and $\mathsf{P}^{s,a}_t$,
> >
> >  over all state-action pairs, i.e. $\frac{1}{|\mathcal{S}||\mathcal{A}|}\sum_{(s,a)}TV(\hat{\mathsf P}^{s,a}, \mathsf{P}^{s,a}_t )$, We reported the average over $20$ runs.
> >
> > We believe you are referring to Figure 3 and Figure 4. We have made the requested changes to the figures.
> >
> > 12.  $\textbf{Typos}$: Thank you for pointing them out! We fixed the typos in the revised version.

---

> > > ### Author Response · Authors · 2024-11-26
> > > **Response to Reviewer Bjbb**
> > >
> > > We sincerely thank the reviewer for taking the time and effort to evaluate our manuscript. If there are any remaining concerns, please do not hesitate to let us know as the discussion period draws to a close.

---

> > > > ### Comment · Reviewer_Bjbb · 2024-11-27
> > > > **Responses to the authors**
> > > >
> > > > I would like to thank the authors for their responses and for revising the manuscript in a short time, I had a better understanding of this submission after the revision and answering my previous questions. Also, I appreciate their effort to improve the manuscript.
> > > >
> > > > ### Responses:
> > > >
> > > > **Contribution 2**: I respectfully disagree with the authors, that their asymptotic proof is based on an unrealistic and infeasible assumption that $n\rightarrow \infty$. Almost any method will converge to the true target estimate given unlimited samples from the target distribution. In short, there is still a big gap between the claim and proof.
> > > >
> > > > **Contribution 3**: **given my limited knowledge driving error bounds** since it depends on the previous contribution, in addition, it is intuitive that ""gap between our policies and the optimal policy in the target domain depends on the total variation distance between the estimated and true target kernel"", since the source and target environments have the same state and action spaces and reward function, but different in the transition kernel.
> > > >
> > > > **Contribution 4**: After checking the revised appendix for the similarities and differences between the source and target transition kernels (and regardless of unnecessary complications and arbitrary values for $(r_s, r_t, \alpha$), the empirical results demonstrate that the estimated target kernel comes close to the true distribution, as the number of the target environment increases. However, the "target task performance" is the average across all state-action pairs, which assumes that all state-action pairs are equally important. Furthermore, the source and target transition kernels are relatively different from each other
> > > >
> > > > Figure 2: still missing the confidence interval.
> > > >
> > > > Figures 2, 3, and 4: the last x-axis stick ($10^4$) is covered by the following sub-figure.
> > > >
> > > > Section 6 (Discussion): This should be integrated into previous sections, allowing a smooth flow of the manuscript.
> > > >
> > > > Given the experiments, I believe this submission is investigating the **transition uncertainty** rather than environmental shift.
> > > >
> > > >
> > > > `In summary, this submission has to scale down its claims or provide concrete evidence supporting claims 2, and 3.`

---

> ### Author Response · Authors · 2024-11-28
> **Response to reviewer Bjbb**
>
> We sincerely appreciate the reviewer's time and effort in reviewing our response and revised manuscript. We will address the additional changes requested and incorporate them into a revised version of the manuscript, which we will post shortly. Below, we provide a detailed, point-by-point response to your additional concerns and questions.
>
> $\textbf{Contribution 2}$: First, we agree with the reviewer that the consistency result, i.e., the pointwise convergence of the MLE, is not surprising given that the MLE is asymptotically consistent under standard regularity conditions on the likelihood function and its derivatives. Accordingly, we will scale down the contribution on the asymptotic consistency of the unconstrained MLE, presenting it as a well-known fact with a concise proof for completeness.
>
> However, it is important to emphasize that, for our purposes, we needed to establish more than consistency—namely, convergence in total variation (TV). TV convergence is a stronger form of convergence, measuring the maximum difference between probabilities assigned to all measurable events by two distributions. Unlike consistency, which ensures pointwise convergence of parameter estimates, TV convergence requires uniform convergence of the distributions, which often necessitates additional conditions.
>
> In our case, because the parameter is a vector of probabilities over a finite state space, we can show that TV convergence follows from consistency. The uniform convergence of the empirical distribution across discrete states ensures that the total difference in probabilities diminishes. We believe that the presentation of the TV convergence result for vanilla IBE (Theorem 8) is essential, as omitting these details would obscure an important aspect of the analysis.
>
> Also, it is important to note that while the consistency of the MLE is well-known, it was crucial to ensure that this property holds under the distance, moment, and density ratio constraints specific to our framework. The consistency of the vanilla IBE (unconstrained MLE) does not inherently extend to the constrained IBE with side information. To address this, we provide concise, rigorous proofs of consistency under each of these constraints, which we consider necessary to establish our findings.
>
> $\textbf{In summary,}$ following the reviewer's comments, we will scale down the contribution on consistency for vanilla IBE, presenting it as a known result with a brief proof for completeness. However, we will retain the TV convergence piece and the short proofs demonstrating that convergence continues to hold in the constrained setting in our framework. We believe this is a balanced approach that avoids repeating known results while doing justice to the new contributions of the paper, maintaining both clarity and rigor.
>
> $\textbf{Contribution 3:}$ Please note that the error bounds we obtained are not straightforward or immediately intuitive, particularly regarding the nature of their dependence on the shift in dynamics. This is especially true in the robust setting, where analyzing the support function over the specified uncertainty set required significant effort. Establishing this result was non-trivial, as it involved bounding the difference between two support functions, which are themselves non-linear optimization problems. This difference had to be carefully analyzed using their dual forms and other advanced techniques, rather than relying on a simple linear difference as in the non-robust setting.
>
> Also, we note that the error bounds established in Theorems 4 and 5 are independent of the previous contribution. Rather, these bounds, together with the convergence of the estimates in TV, lead to the convergence of the evaluation and training errors presented in Theorem 6. We are happy to reframe Theorem 6 as a corollary if desired.
>
> $\textbf{Contribution 4}$:As explained in our previous response, the "target task performance" refers to the average value function, defined as $\frac{1}{|\mathcal{S}|} \sum_{s \in \mathcal{S}} V^{\pi}(s),$
> averaged over 20 runs. This metric evaluates the expected return starting from any state, providing a meaningful measure of performance across the target environment. It is not an average over all state-action pairs.
>
> Regarding the differences between the source and target environments, we acknowledge that they are relatively distinct. Nevertheless, leveraging the side information still yielded significant performance gains, as demonstrated in our experiments. This highlights the robustness of our approach, even in scenarios with substantial domain differences.
>
> While we are not entirely certain about the specific concern you intended to convey in this comment, we are happy to engage in further discussion or clarification.
>
> $\textbf{Figure 2, confidence intervals}$: We have actually included the confidence interval in Figure 2, but it is fairly small. We suggest zooming in on the figure to better view it.

---

> > ### Author Response · Authors · 2024-11-28
> > **Response to reviewer Bjbb**
> >
> > $\textbf{x-tick in Figure 2, 3 and 4 }$: We will fix this in the new revised manuscript.
> >
> > $\textbf{Integrating section 6 to other sections}$: We will integrate the discussion section into previous sections as requested.
> >
> > $\textbf{Our work investigates both the transition uncertainty and environmental shift}$: We clarify that our work investigates both environmental shifts and uncertainty in transition kernels within the target domain. While we assume that both MDPs share the same state and action spaces, we consider two environments with different dynamics.
> >
> > In the non-robust setting, we examine the case where there is a difference in dynamics but no explicit uncertainty set for the target kernel. Specifically, both the training and testing uncertainty sets are singletons, as we set the radius $R=0$. The primary objective in this setting is to achieve superior performance on the target domain compared to vanilla IBE, without explicitly addressing uncertainty in the target domain.
> >
> > In the robust setting, we explicitly account for transition kernel uncertainty by incorporating a specified uncertainty set around the target domain distribution. Our approach seeks policies that are robust across all distributions within this set, ensuring consistent performance despite potential variation in the transition kernel. This approach demonstrates the ability to outperform over-conservative approaches by effectively balancing performance and robustness.
> >
> > $\textbf{Scaling down our claims}$: As requested, we will moderate the claims made in Contribution 2, as detailed in our response above. For Contribution 3, we have clarified that it constitutes a nontrivial advancement, as it is not directly implied by existing results concerning the consistency or convergence properties of the MLE estimator.

---

> > > ### Author Response · Authors · 2024-11-30
> > > **Response to Reviewer Bjbb**
> > >
> > > We have posted a revised version of the paper in which we have addressed the additional concerns. Specifically:
> > > 1) We have $\textbf{scaled down Contribution 2}$, focusing only on the new aspects of convergence in total variation and the handling of the constrained setting. The asymptotic consistency of the MLE is now presented as a known fact.
> > > 2) $\textbf{In the proof of Theorem 1 in Appendix B.1}$, we have removed a previous lemma, which initially provided the MLE estimate in equation (12) along with its proof. Since this lemma follows directly from the consistency of the estimator—a fact we now consider established following the reviewer's suggestion—it is no longer necessary to include it. We now begin the proof by stating:
> > >
> > > "By the consistency of the MLE, for a sequence $x^n = (x_1, \ldots, x_n)$ drawn i.i.d. from the discrete distribution $\mathsf {P}^{s,a}_t$,
> > > the MLE estimate of $\mathsf {P}^{s,a}_t$, is given by
> > >
> > > $\hat{\mathsf P}_{j,n}^{s,a} = \frac{n_j}{n}, \quad j=1, \ldots, k$
> > >
> > > where $n_j := \sum_{i=1}^n \mathbf{1}(x_i = j)$ represents the number of times state $j$ is observed in the sample."
> > >
> > > We have retained only the convergence in total variation in Lemma 8 instead of the former Theorem 8—which needed to be established since it is a stronger form of convergence than pointwise convergence—along with the analysis of convergence in constrained settings. We believe this is a balanced approach that avoids repeating known results while preserving the novel contributions (the stronger form of convergence in total variation that we require and the handling of the constraints).
> > >
> > > 3) We have merged the discussion section into remarks within the relevant sections to improve the flow and coherence of the paper.

---

### Author Response · Authors · 2024-11-23

We would like to thank all three reviewers for their comments, and constructive feedback. We have uploaded a revised version of our manuscript containing all requested changes, highlighted in $\textbf{red}$.  We have also added a Discussion section (section 6), addressing most of the reviewers' concerns.

---

> ### Author Response · Authors · 2024-12-05
>
> We added a revised version of the manuscript in which we add a result that generalizes our previous analysis of the constrained CRB to the vector of moments (section 4.2.2).

---

### Decision · Action_Editor_cj1C · 2025-02-04

**Recommendation:** Reject

**Comment:**

The reviewers recommended decisions were Leaning Accept, Leaning Accept, and Leaning Reject.  None of the reviewers were really arguing that it meets TMLR's criteria and one of the Leaning Accept reviewers even found "the paper does not demonstrate how one would acquire the side information that the algorithm wants to leverage.", finding this to be unaddressed by the authors.

I decided to do a careful read of the revised manuscript.  Sorry, this took so much time.  I give a detailed discussion of my own observations below (although it's a bit more free form than a typical structured review).  My conclusions lined up more closely with Bjbb and in some ways fYMD, but just considered the drawbacks to be more problematic.  Ultimately, I felt like the paper did not deliver clear and convincing evidence of its claims.

I could possibly see the authors submitting a major revision at a later time.  I would recommend they need to either clarify why the results are not simply standard known observations, or tone down the claims that these theoretical results are something new.  Furthermore, they need to develop a realistic scenario of how such constraint-based prior knowledge might arise, and likely even follow through with that example in their empirical results to justify a realistic benefit of the approach.

---

## My Review:

There are two ideas going on that both seem buried, and so it is hard to really recognize if there is some innovation that the TMLR audience would care about.

First, is to note that if t is far enough from s, then building an uncertainty set around MDP s sufficiently large enough to contain t (if somehow you knew how big you would have to make it) could be quite large.  In fact, much larger than the uncertainty set around the MLE t' would have to be to encompass t (if somehow you knew how big it would have to be).  This is an interesting observation, but ultimately it seems to suggest that robust MDP approaches for doing this kind of domain shift are quite poor unless you know the source and target domains have a high degree of similarity.  Placing an uncertainty set around the MLE t' is just offline RL (later you call it Vanila IBE; although I'm not sure what information is being used here to justify the IB part), so the alternative being advocated for isn't really one of transfer or environmental shift at all.

Unless it's motivation depends on the second idea, which I think it does.

The second idea is "information-based" constraints.  The abstract, introduction, preliminaries, and finally approach sections, all keep referring to "information-based estimation", constantly repeating that we could use "prior information", while never saying what form this might take or how you might have it.  Finally we get to Section 4.2, and this information first gets abstracted by some Phi(P_t, P_s) function and still no discussion of what it might look like or where it might come from. Finally at the bottom of page 7 and top of page 8, we get examples of such constraints.  This needs to be frontloaded in the text!  Maybe even in the abstract, but definitely the introduction.  Please give examples of the form of these constraints so the reader doesn't have to read two-thirds of the paper to find that answer.

There are numerous issues in the discussion of these information-based constraints.

Equation (5) doesn't match earlier notation so it's confusing.  Section 3 makes it clear that querying a kernel for the j'th state from s,a would be P^{s,a}(s_j), but here it becomes P^{s,a}_j, which is explained only after it's used.  Should n_j be n^{s,a}_j so as to mean the number of times the next state is s'=j when the previous state is s,a?  This is kind of suggested by the added text prior to equation (5) where it says we explicitly have n samples of the kernel P^{s,a}_t, so I guess we have n samples of kernels from every state-action pair?

Theorem 1 seems to be presented as a significant contribution when I think it barely deserves to be more than an observation.  The assumption of Theorem 1 is that the target kernel is in the constraint set.  That is enough to completely ignore any other details of the constraints.  MLE is a consistent estimator, so given uniform sampling over all s,a pairs, the MLE estimator will converge to the target kernel in the limit (I'm still wrapping my head around whether this guarantees convergence in TV of the parametric distribution, which I think is true since the state space is discrete and finite).  If you assume the global maximum always satisfies the constraints, then the sequence of constrained maximizers is the same as the sequence of unconstrained maximizers.  There's just nothing to prove here.  It remains an interesting observation, but it is stated as if there's some advance, that at the very least is not made clear.

If I'm missing how this is a new result, maybe the text could read, "While we know the MLE estimator will converge to the true transition kernel, we don't know <fill in the blank that the proof concludes beyond this>."

Remark 2 is a very helpful addition.  It touches on the other question I've been hanging on to since the abstract.  Not just what information constraints but where would they come from.  However, there seem to be three examples, and none are that convincing.  TV distance bounds (on individual P^{s,a} kernels no less) between the source and target domain seem possible in the sense that you can use such a bound to say two domains are similar, but how would you have prior knowledge of a numerical value to give.  The second example is density ratio information being inferred or approximated using a small amount of data from the target domain.  Wait!  How?  If I could infer a constraint from a small amount of target domain data, then that constraint would literally not be prior information at all, but rather evidence from data and part of the likelihood.  Finally, the moment constraints seem promising as I can imagine having such constraints since I agree, "in physical systems, aggregate or average behaviors are often measurable, even if the full distribution is unknown."  However the form of this constraint is that the distances between the moments of source and target are constrained.  What about just using the moment knowledge of the target domain as a constraint by itself?  That seems even more powerful than a constraint on the distance of the moments.

In the line before Theorem 4, what is V^*{\cal P}?

How does Theorem 4 bound total-variation of V's?  V is not a probability distribution so it doesn't have a total variation distance.  So what can Theorem 4-6 mean?  If I understand the nature of this contribution, it is saying that if the two transition kernels are close (in TV) than the value functions must be close (probably in max-norm; which is kind of like a TV notion but applies to vectors).  This feels like a well-known result that V^* is smooth with respect to the transition proability matrix (which we can appeal to since the state space is presumed discrete and finite in multiple places of the document).

Let me zoom back out.  The theoretical results are meant to justify the claim that the algorithm is principled and if given suficient data from the target domain it would behave appropriately, converging to the target domain's optimal value function.  This in every way seems true.  Log-likelihood estimates with the assumption that the true target domain satisfies the constraint is sufficient to justify that claim.  That's all that's needed and it is well-known.  But a lot is being made of "proving" this.  Or proving something else that I could not understand.  That seems to suggest the papers claims about what is new is misleading or unclear.

Zooming further out, the first idea really depends on the second as otherwise it's just recommending offline RL (maybe use the data to estimate a model as in Vanilla IBE).  The second idea though is obviously true technically, but lacks a compelling explanation of how such prior knowledge would come about.

The numerical results are maybe the most interesting results in the paper.  I think it's more fair to call Vanilla IBE a baseline than a new result, as it's just doing distributionally robust optimization on the uncertainty set that is the result of fitting the observed data.  This succeeds in that it asymptotically approaches the optimal policy as sample size grows.  Hoever it's often not the best result as at low sample sizes it basically is overly-conservative by necessity (which I'm guessing would result in a random policy).  Some of the other IBE methods can use the closeness constraint to the source domain to do some hopeful transfer.  But isn't this a property of the chosen source and target domains?  There is likely to be many more examples of distant source-target domains where acting random would be a better choice than assuming the source domain is a close approximation of the target.

Overall I'm struggling to see the paper backing up its claims.  And with some of the claims feeling more like well-known observations, I have some doubt about the audience that would gain something from this work.

Going over the rebuttal, I am confused by the comment to reviewer Bjbb about the state-action pairs not being sampled uniformly.  This seems to be contrary to the text of the paper that says that there are n samples from the kernel P_t^{s,a}, which I can ony read as meaning we have n samples for every (s,a).  I think the rest of your comments to the reviewer I agree with, I'm just confused as it seems to contradict the paper.

**Audience:**

I think the issue with the claims also impacts its potential audience.  If the theoretical results are well-established, what's the theoretical findings in the paper that would be of interest to TMLR's audience?  Are the empirical results of interest if it is unclear where the prior knowledge for constraints could come from?  With this said, I don't think potential interest to TMLR's audience is the primary factor in the decision.

**Claims And Evidence:**

The theoretical results do not (at least not clearly) establish a result that is not a textbook result, yet these results are emphasized as significant contributions.  (To be clear, I'm less worried about the motivation and experiments focusing on small samples from a target domain, whereas the theory gives asymptotic results for the infinite sample case.  While obviously finite-sample bounds would be nice, I agree with the authors that this is a useful step in that direction.)  However, I don't think there's a new step being made at all.

The empirical results show an interesting advantage to the constraints, but the claim that such constraints are likely available prior knowledge is not justified.  The added paragraph to discuss this I suppose helps, but doesn't do enough, and the experiments further highlight this in that they don't give a naturalistic means for how the prior knowledge arose to use as constraints.  Suggestions that you can use the target domain data to estimate something like a density ratio and then call this "prior knowledge" is not compelling.

The bulk of the claims are not supported by convincing or clear evidence (theoretical or empirical).

**Resubmission Of Major Revision:**

The authors may consider submitting a major revision at a later time.